# MaMi-HOI: Harmonizing Global Kinematics and Local Geometry for Human-Object Interaction Generation

**Hao Wang** [1]  **Shiqi Wang** [1]  **Qi Liu** [1]

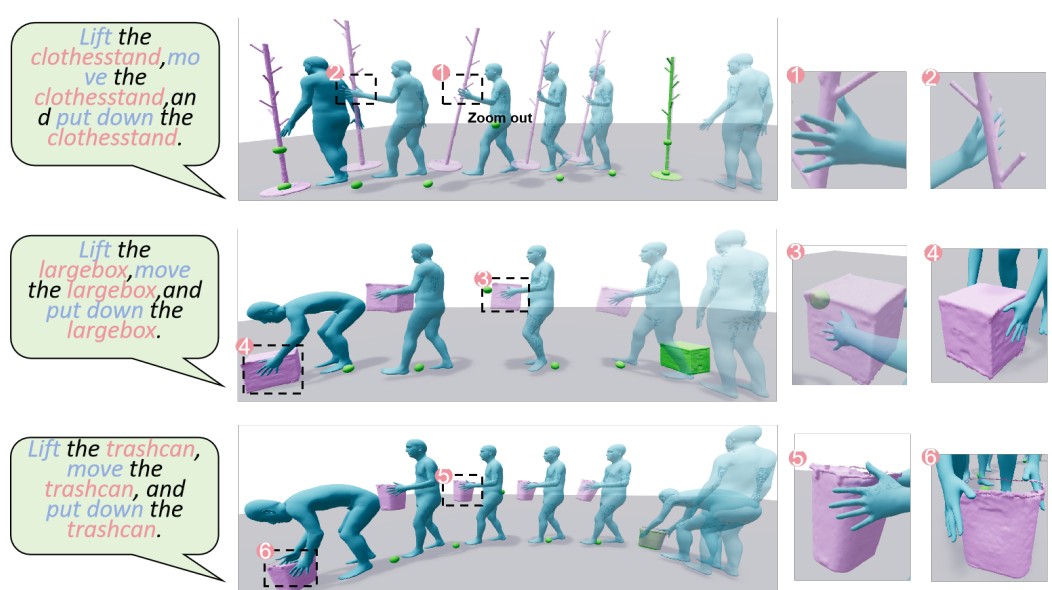

*Figure 1.* High-fidelity interaction synthesis generated by our proposed MaMi-HOI framework. Note the precise contact and natural human motion.

## Abstract

Generating realistic 3D Human-Object Interactions (HOI) is a fundamental task for applications ranging from embodied AI to virtual content creation, which requires harmonizing high-level semantic intent with strict low-level physical constraints. Existing methods excel at semantic alignment, however, they struggle to maintain precise object contact. We reveal a key finding termed *Geometric Forgetting*: as diffusion model depth increases, semantic feature tend to overshadow object geometry feature, causing the model to lose its perception to object geometry. To address this, we propose MaMi-HOI, a hierarchical framework reconciling **Ma**cro-level kinematic fluidity with **Mi**cro-level spatial precision. First, to counteract geometric forgetting, we introduce the Geometry-Aware Proximity Adapter (GAPA), which explicitly re-injects dense object details to perform residual snapping corrections for precise contact. Nevertheless, such aggressive local enforcement can disrupt global dynamics, leading to robotic stiffness. In response, we introduce the Kinematic Harmony Adapter (KHA), which proactively aligns whole-body posture with spatial objectives, ensuring the skeleton actively accommodates constraints without compromising naturalness. Extensive experiments validate that MaMi-HOI simultaneously achieves natural motion and precise contact. Crucially, it extends generation capabilities to long-term tasks with complex trajectories, effectively bridging the gap between global navigation and high-fidelity manipulation in 3D scenes. Code is available at https://github.com/DON738110198/MaMi-HOI.git

[1] School of Future Technology, South China University of Technology, Guangzhou, Guangdong, China. Correspondence to: Qi Liu <drliuqi@scut.edu.cn>.

*Proceedings of the $43^{rd}$ International Conference on Machine Learning*, Seoul, South Korea. PMLR 306, 2026. Copyright 2026 by the author(s).

# 1. Introduction

Synthesizing high-fidelity 3D Human-Object Interactions (HOI) constitutes a cornerstone for advancing embodied artificial intelligence (Guzov et al., 2024), immersive virtual reality (Starke et al., 2019; Xie et al., 2023; Cao et al., 2025), and autonomous robotics (Bharadhwaj et al.). A coherent HOI sequence involves far more than static poses; it requires a continuous, dynamically evolving interplay where an agent must plan its approach, coordinate full-body mechanics to manipulate an object, and maintain precise physical engagement throughout the motion.

Recently, the remarkable success of diffusion models(Ho et al., 2020; Nichol & Dhariwal, 2021; Liu et al., 2024; Ho & Salimans; Dhariwal & Nichol, 2021) has spurred notable efforts to apply them in HOI generation. Despite recent advances(Song et al., 2024; Xie et al., 2024; Cha et al., 2024; Wu et al., 2025a), achieving a balance between kinematic harmony and spatial precision remains a formidable challenge in HOI synthesis. Existing paradigms typically fall into two extremes. One line of work (Li et al., 2024; Xu et al., 2023; Tevet et al., 2022) implicitly entangles human and object states within single-stream diffusion models. While these methods excel at generating diverse semantic behaviors, they frequently struggle to adhere to strict physical constraints, resulting in plausible but spatially misaligned motions such as floating or penetration. Conversely, other strategies (Xie et al., 2024; Xue et al., 2025) rigidly impose spatial constraints via guidance terms. While ensuring proximity, these methods mechanically drag the body towards targets, disrupting the motion manifold and yielding stiff, unnatural dynamics. Consequently, current approaches fail to unify these qualities, generating either natural motions with contact artifacts or precise contacts with robotic stiffness.

A pivotal question in HOI synthesis arises: *Why do existing models struggle to maintain precise spatial alignment, even when provided with explicit object waypoints?* As visualized in Figure 2, our probing experiments offer a domain-specific explanation. We trace the evolution of feature correlations across network layers and observe a distinct trade-off: while the alignment with high-level semantic feature (e.g., the kinematic trajectory for lifting) strengthens in deeper layers, the sensitivity to fine-grained object geometry (e.g., surface boundaries) suffers a catastrophic drop.

This phenomenon, which we term *Geometric Forgetting*, suggests that in standard single-stream diffusion transformers, semantic features tend to overshadow object geometry feature. Consequently, the precise geometric information required for valid contact is diluted as the network focuses on generating coherent global motion, rendering the final output structurally plausible but locally inaccurate.

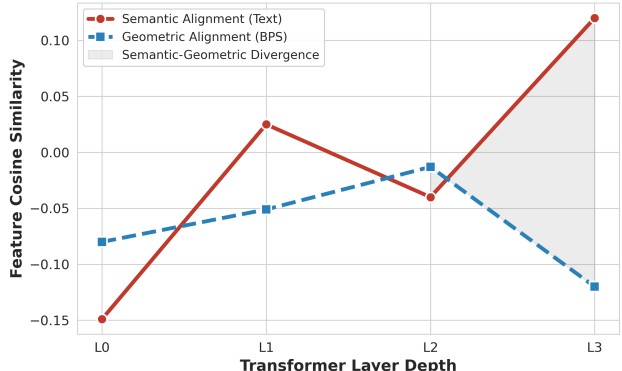

*Figure 2.* Representative Example of Geometric Forgetting. This figure shows the feature cosine similarity between the hidden states of our diffusion transformer and the input conditions for a single interaction sequence. While semantic alignment (red line) increases in deeper layers, geometric alignment (blue line) degrades, illustrating that high-level semantic features can overshadow fine-grained spatial cues. Dataset-wide, statistically robust results and full probing analyses are provided in Appendix E.1.

To break this deadlock, we propose MaMi-HOI, a dual-adapter framework designed to solve the problem. Instead of forcing a single network to balance conflicting goals, we employ a decoupled strategy. To prevent the model from forgetting object geometry, we introduce the Geometry-Aware Proximity Adapter (GAPA). GAPA explicitly retrieves dense geometric details and to ensure this strong geometric injection does not disrupt the natural motion flow, we incorporate the Kinematic Harmony Adapter (KHA). This module actively modulates the generative process, ensuring the whole-body posture adapts coherently to the spatial constraints. MaMi-HOI effectively counteracts geometric forgetting, synthesizing interactions that are both physically accurate and kinematically natural.

Our main contributions are summarized as follows:

- We identify *Geometric Forgetting* as a critical bottleneck in HOI generation, where strong semantic features in deep layers inherently suppress fine-grained spatial cues, preventing existing models from simultaneously achieving semantic fidelity and precise contact.

- We propose MaMi-HOI, which introduces a Retrieve-and-Harmonize paradigm. Instead of simple constraints, it explicitly recovers lost geometric details to enforce contact precision, while synergistically modulating the kinematic chain to maintain whole-body naturalness.

- We demonstrate the effectiveness of our approach through extensive qualitative and quantitative experiments. Crucially, it extends generation capabilities to long-term tasks with complex non-linear trajectories, bridging the gap between global navigation and high-fidelity manipulation in 3D scenes.

## 2. Related Work

### 2.1. Human Object Interaction Generation

Early research primarily addressed interactions with static scenes or objects (Huang et al., 2023; Cen et al., 2024; Wang et al., 2022; Zhao et al., 2023), such as sitting on a sofa. Recently, the field has shifted toward the controllable synthesis of Dynamic HOI, where humans and objects move synchronously (Li et al., 2024; Wu et al., 2025a;b). This evolution enables complex manipulation tasks—such as transporting an object based on instructions—demanding models that handle continuous mutual adaptation. Within this dynamic domain, existing methodologies typically diverge into two streams: joint generation methods (Peng et al., 2025; Diller & Dai, 2024; Zeng et al., 2025; Xue et al., 2025) focus on modeling the coupled distribution of human-object dynamics, utilizing mechanisms like cross-modal attention to improve local correlation. While effective for short-term plausibility, these methods often lack explicit control over global trajectories, limiting their utility in navigation tasks. Conversely, waypoint-guided methods (Li et al., 2023; 2024; Wu et al., 2026) introduce sparse object states as constraints to enable spatial grounding, bridging high-level planning with low-level synthesis. However, a critical limitation persists: these methods typically treat spatial constraints as soft guidance. As network depth increases, strong semantic features tend to overshadow fine-grained spatial cues. Consequently, existing models often forget the precise object shape in deep layers, sacrificing contact accuracy for semantic continuity. Unlike prior works focused on dynamic response (Wu et al., 2026), we target this specific phenomenon to enforce strict geometric adherence.

### 2.2. Geometry-Aware Interaction Modeling

High-fidelity synthesis demands precise micro-level geometry. Traditional architectures rely on standard Cross-Attention, which calculates weights based on semantic similarity rather than physical proximity. This semantic dominance often leads to Ghost Interactions, where distant entities generate false high-weight associations, causing penetration. To mitigate this, recent works have explored explicit geometric modeling. StackFLOW (Huo et al., 2023) utilizes offsets between dense anchors on human-object surfaces to represent spatial relations. CHORE (Xie et al., 2022) predicts a Part Correspondence Field to identify contact interfaces. CONTHO (Nam et al., 2024) estimates vertex-level contact maps to prune erroneous associations, while ROG (Xue et al., 2025) employs an Interaction Distance Field (IDF) to explicitly supervise keypoint distances. Although these explicit constraints improve accuracy, they typically incur high computational costs, require complex post-processing, or rely heavily on auxiliary loss terms rather than inherent network capability. The unresolved

challenge lies in achieving geometric precision efficiently within the generative backbone. To this end, we introduce an architectural inductive bias: distinct from external loss supervision, our method hard-wires distance awareness directly into the attention mechanism, allowing the model to intrinsically perceive and retrieve spatial locality without heavy overhead.

## 3. Method

Given text, object geometry, and sparse waypoints, we aim to synthesize realistic HOI sequences. The core challenge lies in harmonizing synchronized human dynamics with precise physical contact. To address this, **MaMi-HOI** (Figure 3) augments a diffusion backbone with two adapters to decouple macro-dynamics from micro-geometry. The **Kinematic Harmony Adapter (KHA)** operates in parallel to coordinate sparse constraints with the motion manifold, preserving dynamic coherence. Complementarily, the **Geometry-Aware Proximity Adapter (GAPA)** acts as a terminal module, anchoring the pose to dense object geometry for precise contact without disrupting the global structure.

### 3.1. Preliminaries

**Human and Object Motion.** We denote the human motion sequence as $\mathbf{H} \in \mathbb{R}^{T \times D_h}$, where each frame $\mathbf{h}_t$ comprises root translation, linear joint velocities, and 6D continuous rotations (Zhou et al., 2019). We utilize the parametric SMPL-X (Pavlakos et al., 2019) model to reconstruct mesh vertices. For object motion, we define the state sequence as $\mathbf{O} \in \mathbb{R}^{T \times 12}$. Each frame $\mathbf{o}_t$ contains the global centroid and flattened relative rotation matrix. Vertices are recovered via rigid transformation $\mathcal{V}_t = R_{rel}\mathcal{V} + \mathbf{t}_{global}$ with respect to the canonical geometry $\mathcal{V}$.

**Object Geometry (BPS).** To capture fine-grained geometric details, we adopt the Basis Point Set (BPS) (Prokudin et al., 2019) representation. We sample $N = 1024$ fixed points within a unit sphere and compute directional vectors to their nearest neighbors on the object surface. This yields a dense feature tensor $\mathcal{G} \in \mathbb{R}^{N \times 3}$, encoding the precise spatial relationship between the environmental basis and the object geometry.

**Input Condition Representation.** We integrate three conditions to guide diffusion. First, the text $\mathcal{T}$ is encoded by a frozen CLIP (Radford et al., 2021) into a global embedding $c_{text}$ for semantic guidance. Second, spatial guidance is provided by sparse waypoints, formulated as a masked sequence $\mathcal{S} \in \mathbb{R}^{T \times (12 + D_h)}$ (Li et al., 2023). This encodes 2D object positions every 30 frames and a final 3D position, with unconstrained frames zero-padded. Finally, raw BPS features $\mathcal{G}$ are projected and broadcasted to form the

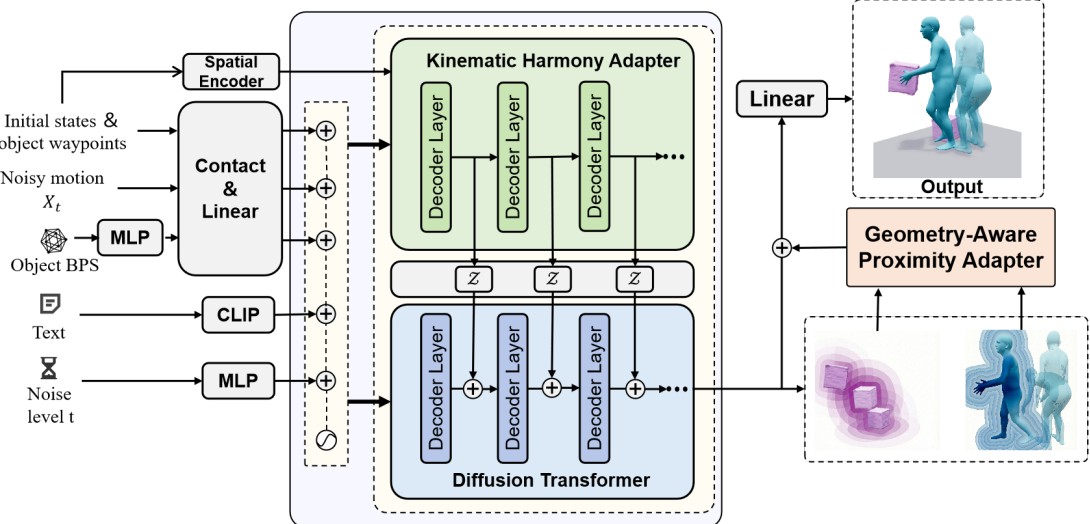

*Figure 3.* **Overview of MaMi-HOI.** Our framework employs a Dual-Adapter paradigm to reconcile motion naturalness and contact precision. The Kinematic Harmony Adapter acts as a trainable copy to maintain global pose coherence, while the Geometry-Aware Proximity Adapter serves as a terminal module to explicitly inject local geometric constraints for accurate contact alignment.

geometry embedding $\hat{\mathcal{G}} \in \mathbb{R}^{T \times 256}$, utilized for fine-grained contact modeling.

**Conditional Diffusion Model** We formulate the unified human-object motion generation as a conditional denoising process. Let $\mathbf{x}_0 = [\mathbf{H}, \mathbf{O}]$ denote the clean joint state of the human and object. Following the standard formulation of Denoising Diffusion Probabilistic Models (DDPM) (Ho et al., 2020), the framework consists of a forward process and a reverse process. The forward process is a fixed Markov chain that gradually injects Gaussian noise into the clean motion $\mathbf{x}_0$ according to a variance schedule $\{\beta_t \in (0,1)\}_{t=1}^{T}$. At any arbitrary timestep $t$, the noisy latent state $\mathbf{x}_t$ can be sampled directly from $\mathbf{x}_0$ via the following closed-form distribution:

$$q(\mathbf{x}_t|\mathbf{x}_0) = \mathcal{N}(\mathbf{x}_t; \sqrt{\bar{\alpha}_t}\mathbf{x}_0, (1 - \bar{\alpha}_t)\mathbf{I}), \qquad (1)$$

where $\alpha_t = 1 - \beta_t$ and $\bar{\alpha}_t = \prod_{s=1}^{t} \alpha_s$. As $t \to T$, the distribution of $\mathbf{x}_T$ approximates a standard isotropic Gaussian $\mathcal{N}(\mathbf{0}, \mathbf{I})$. The generative process is defined as the reverse Markov chain, where the model learns to recover the clean data $\mathbf{x}_0$ from pure noise $\mathbf{x}_T$ conditioned on the multi-modal guidance $\mathcal{C}$. This transition is parameterized by a denoiser network $\varepsilon_\theta$:

$$p_\theta(\mathbf{x}_{t-1}|\mathbf{x}_t, \mathcal{C}) = \mathcal{N}(\mathbf{x}_{t-1}; \mu_\theta(\mathbf{x}_t, t, \mathcal{C}), \Sigma_\theta(\mathbf{x}_t, t, \mathcal{C})), \quad (2)$$

In our framework, the conditioning set $\mathcal{C} = \{c_{text}, \mathcal{S}, \hat{\mathcal{G}}\}$ integrates semantic, kinematic, and geometric signals.

The network is trained to predict the added noise $\epsilon$ by minimizing the simple mean squared error (MSE) objective:

$$\mathcal{L} = \mathbb{E}_{\mathbf{x}_0, t, \mathcal{C}, \epsilon \sim \mathcal{N}(\mathbf{0}, \mathbf{I})} \left[ \|\epsilon - \varepsilon_\theta(\mathbf{x}_t, t, \mathcal{C})\|^2 \right]. \qquad (3)$$

During inference, we iteratively denoise sampled Gaussian noise using the predicted $\varepsilon_\theta(\mathbf{x}_t, t, \mathcal{C})$ to synthesize the final motion sequence.

### 3.2. Geometry-Aware Proximity Adapter (GAPA)

To counteract the *Geometric Dilution* phenomenon where deep diffusion layers lose sensitivity to spatial cues, we introduce the Geometry-Aware Proximity Adapter (GAPA). Functioning as a terminal retrieval mechanism, GAPA explicitly re-injects the forgotten dense object geometry $\mathcal{G}$ (BPS) into the semantic-heavy motion representations. We formulate this process as a Distance-Aware Cross-Attention module (Figure 4) that guides precise contact via residual geometric corrections. Inspired by the vector attention mechanism in point cloud processing (Zhao et al., 2021), we formulate GAPA as a Distance-Aware Cross-Attention module. This design explicitly injects geometric locality into the semantic reasoning process.

**Geometric Feature Projection.** We first establish a shared coordinate frame by projecting the motion features $\mathbf{F}_{mot}$ and object BPS embeddings $\mathbf{F}_{obj}$ into a latent geometric space via learnable projectors $\phi_m$ and $\phi_o$, yielding coordinate representations $\mathbf{P}_{mot}$ and $\mathbf{P}_{obj}$ respectively.

**Distance-Biased Attention.** To enforce strict spatial locality, we compute the relative difference vector $\boldsymbol{\delta}_{ij} = \mathbf{P}_{mot}^{(i)} - \mathbf{P}_{obj}^{(j)}$ for each feature pair. We then incorporate the squared Euclidean distance $\|\boldsymbol{\delta}_{ij}\|^2$ as a negative bias into the attention scoring function:

$$\mathbf{S}_{ij} = \frac{\mathbf{Q}_i \mathbf{K}_j^\top}{\sqrt{d}} - \gamma \cdot \|\boldsymbol{\delta}_{ij}\|^2, \qquad (4)$$

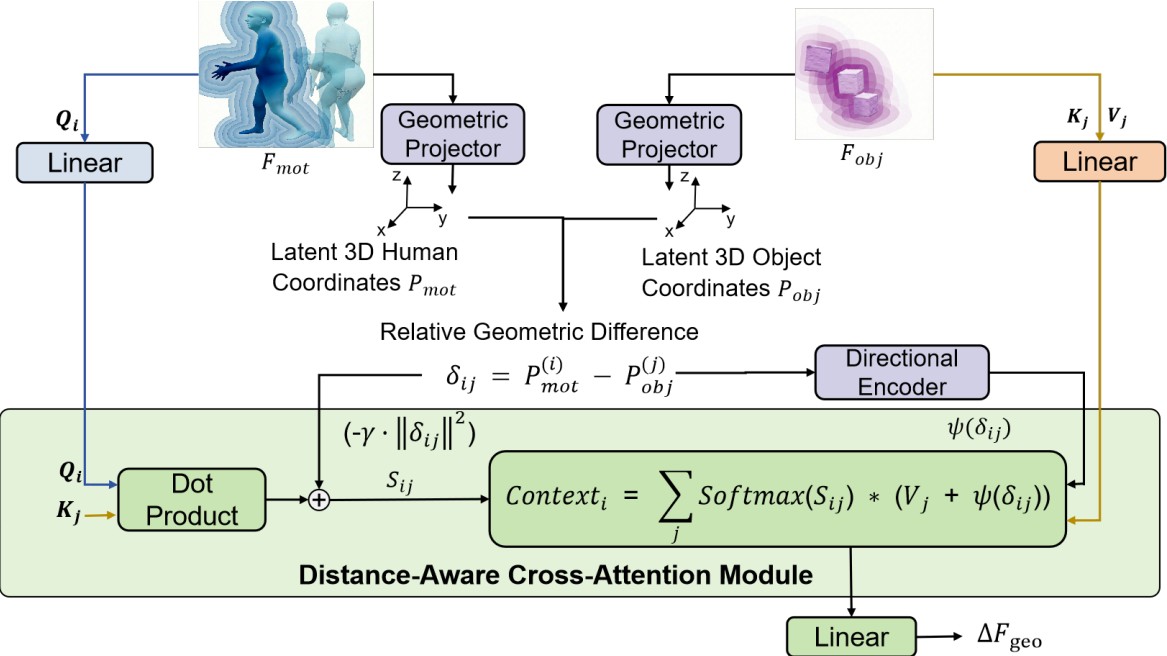

*Figure 4.* **Architecture of GAPA.** This module recovers fine-grained spatial cues via distance-aware cross-attention. It computes relative geometric context from object BPS features and injects it as a residual correction to the motion stream.

where $\gamma$ is a learnable scalar controlling sensitivity to physical proximity. This formulation naturally suppresses irrelevant long-range interactions, focusing the model's attention solely on the immediate contact geometry.

**Residual Snapping Correction.** We explicitly fuse relative spatial information into the value aggregation to compute the geometric context:

$$\mathbf{Context}_i = \sum_j \text{Softmax}(\mathbf{S}_{ij}) \cdot (\mathbf{V}_j + \psi(\boldsymbol{\delta}_{ij})). \quad (5)$$

Crucially, this context is projected to form a residual field $\Delta\mathbf{F}_{geo}$ and added to the original features: $\mathbf{F}_{refined} = \mathbf{F}_{mot} + \Delta\mathbf{F}_{geo}$. This residual formulation ensures GAPA operates as a non-destructive refinement, preserving the semantic motion structure while applying precise additive adjustments to snap end-effectors onto the object surface.

Distinct from recent works like ROG (Xue et al., 2025) that rely on implicit interactive distance fields which may suffer from ambiguity, or CHOIS (Li et al., 2024) that enforce contact via rote memorization of loss terms, GAPA architecturally hard-wires an explicit geometric inductive bias into the network. By directly injecting relative spatial vectors, it provides deterministic and fine-grained correction cues that generalize better to unseen shapes. However, such aggressive local enforcement can disrupt the global motion manifold, potentially causing artifacts like limb overstretching. We address this side effect using the kinematic harmonization mechanism introduced next.

### 3.3. Kinematic Harmony Adapter (KHA)

While GAPA effectively retrieves lost geometry to enforce contact, its aggressive local snapping can compromise motion naturalness. To reconcile rigid geometric constraints with fluid dynamics, we introduce the Kinematic Harmony Adapter (KHA). Unlike post-processing, KHA acts as a parallel resonance module that implicitly modulates the generative flow, ensuring the skeleton actively accommodates spatial objectives rather than being passively deformed.

**Spatial-Aware Copy Mechanism.** We first encode the sparse waypoints $\mathcal{S}$ via an MLP and fuse them with the sinusoidal timestep embedding. Structurally, KHA mirrors the architecture of the main Diffusion Transformer as a parallel branch. It accepts the same motion inputs but is distinctively conditioned on the spatial embedding. This design allows KHA to specialize in extracting control-specific features to enforce trajectory adherence, while the main backbone focuses on modeling the intrinsic motion distribution.

**Zero-Linear Injection.** To seamlessly transfer learned constraints, we connect each KHA layer to the corresponding main branch layer via a zero-initialized linear layer $\mathcal{Z}(\cdot)$. The injection is formulated as:

$$\mathbf{F}_{main}^{(l)} \leftarrow \mathbf{F}_{main}^{(l)} + \mathcal{Z}(\mathbf{F}_{kha}^{(l)}). \quad (6)$$

Initialized to output null signals, this mechanism ensures the model retains its original distribution at the start of training. As training progresses, KHA learns to inject meaningful spatial modulations, guiding the generated features to

*Table 1.* **Comparison of methods on the FullBodyManipulation dataset.** Arrows indicate whether lower (↓) or higher (↑) is better. Best results are highlighted in **bold**. Note that MaMi-HOI achieves the best balance between contact precision and motion naturalness.

| Method | Condition Matching | | | Human Motion | | | | Interaction | | | | | GT Difference | | | |
|---|---|---|---|---|---|---|---|---|---|---|---|---|---|---|---|---|
| | $T_s \downarrow$ | $T_e \downarrow$ | $T_{xy} \downarrow$ | $H_{feet} \downarrow$ | FS $\downarrow$ | $R_{prec} \uparrow$ | FID $\downarrow$ | $C_{prec} \uparrow$ | $C_{rec} \uparrow$ | $C_{F1} \uparrow$ | C% $\uparrow$ | $P_{hand} \downarrow$ | MPJPE $\downarrow$ | $T_{root} \downarrow$ | $T_{obj} \downarrow$ | $R_{obj} \downarrow$ |
| InterDiff | 0.00 | 158.84 | 72.72 | **0.90** | 0.42 | 0.08 | 208.0 | 0.63 | 0.28 | 0.33 | 0.27 | 0.55 | 25.91 | 63.44 | 88.35 | 1.65 |
| MDM | 5.18 | 33.07 | 19.42 | 6.72 | 0.48 | 0.51 | 6.16 | 0.72 | 0.47 | 0.53 | 0.43 | 0.66 | 17.86 | 34.16 | 24.46 | 1.85 |
| Lin-OMOMO | 0.00 | 0.00 | 0.00 | 7.21 | 0.41 | 0.29 | 15.33 | 0.68 | 0.56 | 0.57 | 0.54 | **0.51** | 21.73 | 36.62 | 17.12 | 1.21 |
| Pred-OMOMO | 2.39 | 8.03 | 4.15 | 7.08 | 0.40 | 0.54 | 4.19 | 0.73 | 0.66 | 0.66 | 0.62 | 0.58 | 18.66 | 28.39 | 16.36 | 1.05 |
| GT-OMOMO | 0.00 | 0.00 | 0.00 | 7.10 | 0.41 | 0.48 | 5.69 | 0.77 | 0.66 | 0.67 | 0.59 | 0.55 | 15.82 | 24.75 | 0.00 | 0.00 |
| CHOIS | 1.71 | 6.31 | 2.87 | 4.20 | **0.35** | 0.64 | 0.69 | 0.80 | 0.64 | 0.67 | 0.54 | 0.59 | 15.30 | 24.33 | 12.53 | 0.99 |
| HOI-Dyn | 1.75 | 5.58 | 3.26 | 3.07 | 0.37 | - | 0.48 | - | - | 0.71 | 0.60 | 0.64 | 15.60 | 23.90 | 12.47 | 0.90 |
| MaMi-HOI (Ours) | **1.53** | **4.51** | **2.14** | 4.10 | 0.37 | **0.71** | **0.39** | **0.80** | **0.72** | **0.73** | **0.62** | 0.57 | **13.50** | **19.65** | **9.98** | **0.78** |

*Table 2.* **Interaction synthesis results on the 3D-FUTURE dataset.** This table evaluates the generalization capability to unseen object geometries. Despite not being trained on these shapes, MaMi-HOI maintains superior motion quality and contact fidelity.

| Method | Condition Matching | | | Human Motion | | | | Interaction | |
|---|---|---|---|---|---|---|---|---|---|
| | $T_s \downarrow$ | $T_e \downarrow$ | $T_{xy} \downarrow$ | $H_{feet} \downarrow$ | FS $\downarrow$ | $R_{prec} \uparrow$ | FID $\downarrow$ | C% $\uparrow$ | $P_{hand} \downarrow$ |
| InterDiff | 0.00 | 161.26 | 72.77 | -0.26 | 0.42 | 0.09 | 207.3 | 0.24 | **0.11** |
| MDM | 12.58 | 40.55 | 28.72 | 7.02 | 0.49 | 0.53 | 8.50 | 0.34 | 0.26 |
| Lin-OMOMO | 0.00 | 0.00 | 0.00 | 6.32 | 0.42 | 0.23 | 23.17 | 0.44 | 0.11 |
| Pred-OMOMO | 4.15 | 9.03 | 3.89 | 6.08 | 0.40 | 0.46 | 3.74 | 0.50 | 0.18 |
| CHOIS | 4.12 | 7.35 | 2.92 | 3.75 | 0.38 | 0.62 | 1.67 | 0.47 | 0.19 |
| HOI-Dyn | 4.60 | 6.17 | 2.95 | **2.56** | 0.37 | - | 1.62 | 0.54 | 0.26 |
| MaMi-HOI (Ours) | **3.43** | **5.87** | **2.21** | 3.35 | **0.37** | **0.69** | **1.57** | **0.54** | 0.23 |

satisfy waypoints while preserving the intrinsic kinematic coherence of the original data manifold.This mechanism implicitly modulates the generated features, ensuring the final motion naturally satisfies spatial constraints while preserving the kinematic coherence of the original data manifold.

### 3.4. Synergistic Integration

MaMi-HOI unifies two complementary mechanisms to resolve the conflict between geometric retrieval and kinematic preservation. From a mechanistic perspective, GAPA functions as an explicit geometric retrieval module, performing attention-based snapping to guarantee sub-centimeter contact precision. However, such local enforcement operates independently of global dynamics. Complementarily, KHA acts as an implicit kinematic modulator. By proactively aligning the global pose distribution through the trainable copy, KHA ensures the whole body anticipates the contact. This establishes a coarse-to-fine synergy: KHA establishes a kinematically plausible global posture, minimizing the deformation burden for GAPA's terminal fine-tuning, thereby effectively balancing physical accuracy with biological naturalness.

## 4. Experiments

### 4.1. Experimental Setup

**Datasets.** Following CHOIS (Li et al., 2024), we evaluate on: (i) **FullBodyManipulation** (Li et al., 2024): Contains ~10 hours of paired MoCap data across 15 object categories.

We use the standard split with 15 subjects for training and 2 for testing. (ii) **3D-FUTURE** (Fu et al., 2021): We use 17 unseen furniture models paired with test trajectories to assess generalization to novel geometries.

**Metrics.** We assess performance across four dimensions: (i) **Matching**: Euclidean errors for start ($T_s$), end ($T_e$), and intermediate ($T_{xy}$) waypoints. (ii) **Motion Quality**: Foot sliding ($FS$), foot height ($H_{feet}$), FID, and R-precision ($R_{prec}$). (iii) **Interaction Quality**: Contact precision/recall/F1 ($C_{prec}, C_{rec}, C_{F1}$), contact ratio ($C_\%$), and SDF-based penetration ($P_{hand}$). (iv) **Reconstruction**: MPJPE, and translation/rotation errors for root and object ($T_{root/obj}, O_{root/obj}$).

**Implementation.** We use a frozen CLIP (ViT-B/32) encoder. The model is trained via Adam ($lr = 2 \times 10^{-4}$) for 400k steps (batch size 32) with 1,000 diffusion steps on a single RTX 3090 GPU.

**Baselines.** We compare MaMi-HOI against SOTA methods CHOIS (Li et al., 2024) and HOI-Dyn (**?**). Additionally, we reference adapted baselines including InterDiff (Xu et al., 2023), MDM (Tevet et al., 2022), and OMOMO variants (Li et al., 2023) as reported in (Li et al., 2024).

### 4.2. Quantitative Evaluation

We first evaluate generation quality on the FullBodyManipulation dataset. As shown in Table 1, MaMi-HOI achieves state-of-the-art performance, effectively balancing spatial precision with motion fidelity.

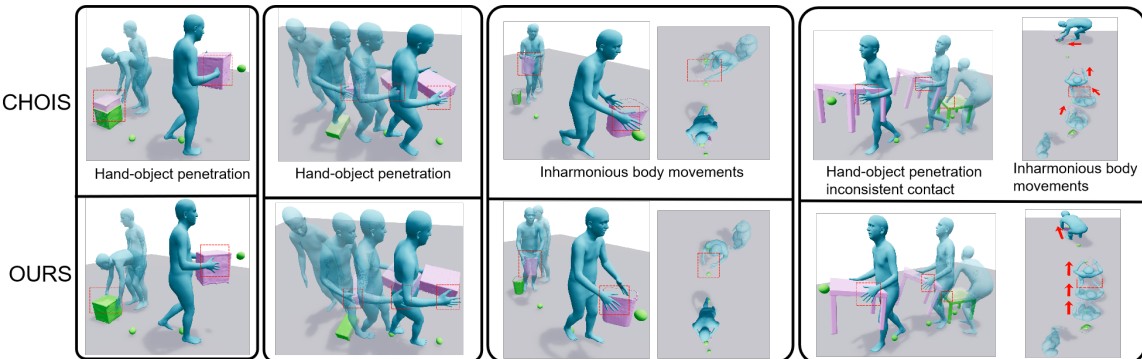

*Figure 5.* Qualitative comparisons. We visualize sample sequences generated by CHOIS and our MaMi-HOI. As highlighted by the red dashed boxes, CHOIS frequently suffers from severe physical violations, including *hand-object penetration*, *floating artifacts*, and *inharmonious body movements*. In contrast, MaMi-HOI generates physically plausible interactions with precise surface contact and coherent whole-body dynamics, effectively mitigating these artifacts.

In Condition Matching, MaMi-HOI yields the lowest errors $(T_s, T_e, T_{xy})$, significantly outperforming HOI-Dyn. This confirms that our dual-adapter framework strictly enforces adherence to input waypoints. Regarding Motion Quality, we attain the best FID and R-precision, surpassing CHOIS. This indicates our generated motions are both distributionally closer to real dynamics and semantically consistent. Notably, the superior FID suggests KHA harmonizes the global manifold without compromising stability.

Crucially, in Interaction Quality, MaMi-HOI achieves the highest F1 score and Contact Percentage. Unlike prior methods where improved contact often worsens interpenetration, we maintain a lower Penetration Score than both CHOIS and HOI-Dyn. This high contact, low penetration profile validates GAPA's ability to resolve fine-grained physical conflicts.

Table 2 reports the performance on the 3D-FUTURE dataset, where the model must generalize to unseen object geometries. MaMi-HOI maintains its superiority in spatial alignment, achieving the lowest Condition Matching errors. Significantly, our method generates much more realistic motions compared to MDM and Lin-OMOMO, and surpasses CHOIS. While our penetration score is slightly higher than CHOIS, we achieve a noticeably higher Contact Percentage and $R_{prec}$. This suggests that baselines like CHOIS may minimize penetration by avoiding contact on complex unseen shapes, whereas MaMi-HOI actively attempts to satisfy contact constraints, resulting in a more plausible overall interaction despite the challenging domain gap. Quantitative analysis of hand-object alignment on unseen object geometries is included in Appendix E.6.

### 4.3. Qualitative Results

Figure 5 visualizes the comparison between MaMi-HOI and CHOIS (Li et al., 2024). As observed, CHOIS frequently

suffers from severe hand-object penetration or floating artifacts due to its reliance on soft constraints. Additionally, it often yields inharmonious body dynamics, such as unnatural spine curvature, indicating a disconnection between local goals and global kinematics.

In contrast, MaMi-HOI demonstrates superior fidelity. Driven by GAPA, our model eliminates interpenetration and ensures stable contact, while KHA simultaneously harmonizes the global posture to produce coherent motion. This visual evidence confirms that our dual-adapter framework successfully reconciles geometric precision with kinematic naturalness.

### 4.4. Ablation Studies

To assess the contribution of each component in our proposed MaMi-HOI, we conduct an ablation study on the FullBodyManipulation dataset by individually removing the Kinematic Harmony Adapter (KHA) and the Geometry-Aware Proximity Adapter (GAPA). The results are summarized in Table 3. Removing KHA leads to a notable degradation in human motion quality, evidenced by a significant increase in FID and larger ground truth differences, confirming its role in ensuring global kinematic coherence and realism. Conversely, removing GAPA primarily impairs spatial precision, resulting in sharply increased condition matching errors and slightly higher hand-object penetration. This demonstrates GAPA's criticality for fine-grained geometric alignment. The full MaMi-HOI model achieves the optimal balance across all metrics, underscoring the complementary nature of both adapters in generating interactions that are both physically precise and kinematically natural.

### 4.5. Application in 3D Scenes

To demonstrate robust applicability, we deploy MaMi-HOI in complex 3D environments using the Replica

*Table 3.* **Ablation study** on the FullBodyManipulation dataset. We measure the effect of different module in the human and object motion generation.

| Method | Condition Matching | | | Human Motion | | | | GT Difference | | | |
|---|---|---|---|---|---|---|---|---|---|---|---|
| | $T_s \downarrow$ | $T_e \downarrow$ | $T_{xy} \downarrow$ | $H_{feet} \downarrow$ | FS $\downarrow$ | $R_{prec} \uparrow$ | FID $\downarrow$ | MPJPE $\downarrow$ | $T_{root} \downarrow$ | $T_{obj} \downarrow$ | $O_{obj} \downarrow$ |
| MaMi-HOI w/o KHA | 1.76 | 5.89 | 2.70 | 4.05 | 0.39 | 0.70 | 0.53 | 13.60 | 20.34 | 10.57 | 0.80 |
| MaMi-HOI w/o GAPA | 1.69 | 5.89 | 2.51 | **3.89** | 0.37 | 0.71 | **0.36** | 14.31 | 21.02 | 11.41 | 0.82 |
| MaMi-HOI (ours) | **1.53** | **4.51** | **2.14** | 4.10 | **0.37** | **0.71** | 0.39 | **13.53** | **19.65** | **9.98** | **0.78** |

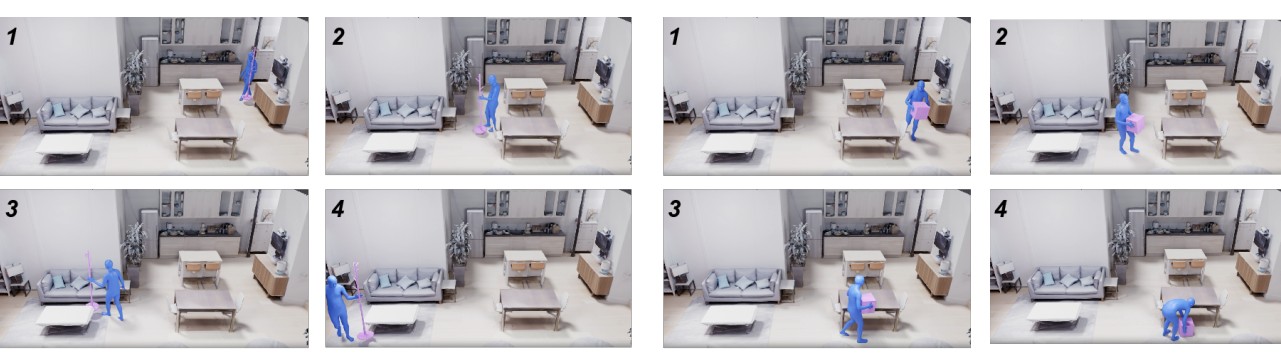

(a) Interaction with a clothesstand  (b) Interaction with a large box

*Figure 6.* Application in realistic 3D scenes. We visualize the synthesized interactions within a scene from the Replica dataset, conditioned on text instructions. These examples demonstrate MaMi-HOI's capability to generate environment-conscious and physically plausible motions that adapt to diverse object geometries.

dataset (Straub et al., 2019). The integration follows a three-stage pipeline: (i) instruction parsing identifies target objects and semantics from text commands; (ii) global planning utilizes Habitat to compute collision-free navigation paths; and (iii) interaction synthesis employs MaMi-HOI to generate fine-grained manipulation motions upon reaching the interaction zone.

Figure 6 illustrates coherent agent behaviors in cluttered rooms. In case (a), the agent naturally drags a clothes stand, adapting its posture to the object structure. In case (b), it performs a coordinated full-body push on a large box. These examples highlight MaMi-HOI's ability to seamlessly bridge global navigation with high-fidelity local interactions for embodied applications.

*Table 4.* **Long-term interaction synthesis results** on the FullBodyManipulation and 3D-FUTURE datasets. * represents the results on the 3D-FUTURE dataset.

| | Condition Matching | | | Human Motion | | Interaction |
|---|---|---|---|---|---|---|
| | $T_s \downarrow$ | $T_e \downarrow$ | $T_{xy} \downarrow$ | $H_{feet} \downarrow$ | FS $\downarrow$ | $C_\% \uparrow$ |
| CHOIS | 2.22 | 9.94 | 5.73 | 4.57 | 0.46 | 0.63 |
| MaMi-HOI (ours) | **1.05** | **4.04** | **2.95** | **4.31** | **0.48** | **0.79** |
| CHOIS* | 5.62 | 12.08 | 5.95 | 4.27 | **0.41** | **0.65** |
| MaMi-HOI* (ours) | **2.41** | **7.75** | **3.17** | **3.95** | 0.48 | 0.53 |

**Quantitative Evaluation of Long-term Synthesis.** To further validate the reliability of our method in complex scenarios like the 3D scene application, we conduct a quan-

titative evaluation on long-term interaction synthesis. As reported in Table 4, MaMi-HOI significantly outperforms the baseline CHOIS on both the FullBodyManipulation and 3D-FUTURE datasets. Notably, our method reduces the trajectory end-point error ($T_e$) by over 50%, ensuring that the agent accurately reaches the target object after a long navigation sequence. Furthermore, MaMi-HOI maintains a high Contact Percentage ($C_\%$), confirming that the generated interactions remain physically effective even over extended temporal horizons.Additional task-specific evaluations, including per-module effects and long-term generalization, are included in Appendix E.5.

## 5. Conclusion

This work addresses the phenomenon of *Geometric Forgetting*, revealing that relying solely on implicit feature entanglement is insufficient for high-fidelity HOI generation. Through the proposed MaMi-HOI framework, we demonstrate that explicitly injecting lost geometric details is essential for harmonizing fluid whole-body dynamics with strict physical constraints. Furthermore, our results in long-term synthesis serve as a promising step toward integrating global movement with fine-grained manipulation in realistic environments.

Current limitations stem primarily from data and representation constraints. Reliance on standard SMPL-X topology and existing datasets restricts finger-level resolution, poten-

tially omitting high-frequency articulatory details despite accurate contact. Future work aims to integrate high-fidelity hand models and explore multi-agent scalability.

## Impact Statement

This paper presents work whose goal is to advance the field of Machine Learning. Our proposed framework, MaMi-HOI, enhances the realism and physical plausibility of 3D human-object interaction synthesis. This advancement has positive implications for downstream applications such as virtual reality, gaming, and synthetic data generation for embodied AI, potentially reducing the manual effort required for high-fidelity animation. While generative models carry inherent risks related to the creation of misleading content, our work focuses on generic interaction physics rather than identity-specific synthesis. To the best of our knowledge, there are no specific negative societal consequences that must be highlighted here.

## Acknowledgments

This work was supported in part by the GJYC program of Guangzhou under Grant 2024D01J0081, in part by the ZJ program of Guangdong under Grant 2023QN10X455, and in part by the Fundamental Research Funds for the Central Universities under Grant 2025ZYGXZR053. Sponsored by CAAI-Lenovo Blue Sky Research Fund.

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

# A. Appendix

In this appendix, we provide additional implementation details and experimental results to support the findings in the main paper. The content is organized as follows:

- **Section B**: Detailed definitions of the evaluation metrics.

- **Section C**: Detailed specifications of motion and condition representations.

- **Section D**: Hyperparameters and architectural details of the Backbone.

- **Section E**: Additional diagnostic analyses, module ablations, efficiency measurements, and downstream evaluations.

- **Section F**: Additional qualitative results demonstrating generalization and diverse interaction scenarios.

# B. Evaluation Metrics Details

To ensure a comprehensive and rigorous assessment, we adopt a multi-dimensional evaluation protocol covering condition adherence, motion quality, interaction fidelity, and reconstruction accuracy. Since standard feature extractors for generic human motion (e.g., T2M) do not account for object states, we first describe the customized feature extraction backbone used for our perceptual metrics.

### B.1. Text and HOI Feature Extraction

Following the evaluation protocols established in CHOIS (Li et al., 2024), we employ a specialized feature extractor designed to assess Human-Object Interaction (HOI) motions. This framework extends the standard Text-to-Motion (T2M) evaluation backbone (Guo et al., 2022) to jointly encode human and object dynamics, enabling the computation of perceptual metrics such as FID and R-precision.

**Architecture.** The evaluation model consists of two co-embedding streams:

- **Text Stream:** A frozen CLIP (ViT-B/32) text encoder is used to project textual descriptions into a semantic feature space.

- **Motion Stream:** A bidirectional GRU (BiGRU) network processes the generated HOI sequences to extract kinematic features.

**Input Representation.** As defined in the CHOIS framework, we modify the input dimensions of the BiGRU to explicitly incorporate object states alongside human poses. The motion representation vector for each frame has a length of 216, allocated as follows:

- **Human Joints (72 dims):** Representing the 3D positions of the human's 24 joints ($24 \times 3$).

- **Human Rotations (132 dims):** Representing the 6D continuous rotation features for the 22 body joints ($22 \times 6$).

- **Object Rotation (9 dims):** The flattened $3 \times 3$ relative rotation matrix of the object.

- **Object Transformation (3 dims):** The global translation vector of the object.

**Training Objective.** The motion encoder is trained using a contrastive loss to minimize the feature distance between matched text-HOI pairs while maximizing the distance between mismatched pairs. This establishes a strong alignment between textual semantics and HOI kinematics.

**Training Split and Leakage Control.** To ensure a fair evaluation, the HOI-BiGRU evaluator is trained exclusively on the training split of the FullBodyManipulation dataset, following the CHOIS evaluation protocol. FullBodyManipulation uses a strict subject split with 15 subjects for training and 2 subjects for testing, and all reported metrics are computed on unseen test motions. For the 3D-FUTURE evaluation, we pair 17 unseen furniture models with motions from the FullBodyManipulation test split; therefore, the evaluator has not observed either the test motions or the novel object geometries during its training.

## B.2. Condition Matching Metrics

These metrics quantify the controllability of our model by measuring the deviation between the generated object trajectory and the input sparse waypoints.

- **Start/End Error** ($T_s, T_e$)**:** The Euclidean distance (in cm) between the generated object position and the ground truth at the first frame ($t = 0$) and the last frame ($t = T$).

- **Waypoint Error** ($T_{xy}$)**:** The average Euclidean distance (in cm) measured at all intermediate constrained keyframes (e.g., every 30 frames). This metric reflects the model's ability to adhere to the spatial guidance throughout the sequence.

## B.3. Human Motion Quality Metrics

We assess the physical plausibility and semantic consistency of the generated human motion using the following indicators:

- **Foot Sliding (FS):** Following prior work (Li et al., 2023), foot sliding is defined as the accumulation of horizontal translation when the foot should be planted. Specifically, it is calculated as the weighted average of the foot's displacement in the $xy$-plane when the foot height is below a threshold (2.5 cm) and the velocity exceeds a tolerance limit. The unit is centimeters (cm).

- **Foot Height** ($H_{feet}$)**:** This metric measures the average height of the feet relative to the ground plane (in cm), serving as a check for artifacts such as floor penetration or unnatural floating.

- **Fréchet Inception Distance (FID):** FID assesses the distribution distance between the generated motions and the real ground-truth motions. We compute FID using the features extracted by HOI-BiGRU model described in Section B. A lower FID indicates that the generated distribution is closer to the real data distribution.

- **R-precision** ($R_{prec}$)**:** We report the Top-3 R-precision. For each generated motion sequence, we retrieve the top-3 most similar text descriptions from a pool of 32 candidates (1 matched + 31 mismatched) based on the Euclidean distance in the joint feature space. A higher R-precision indicates better semantic consistency between the motion and the input text.

## B.4. Interaction Quality Metrics

High-fidelity HOI synthesis requires precise hand-object spatial relations. We evaluate this using both contact and penetration metrics.

- **Contact Accuracy** ($C_{prec}, C_{rec}, C_{F1}$)**:** Following (Li et al., 2023), we define a valid contact if the distance between any object vertex and the nearest human hand vertex is less than a threshold $\tau = 4$ mm. Based on this definition, we compute:
  - **Precision** ($C_{prec}$)**:** The ratio of correctly predicted contact frames to the total predicted contact frames.
  - **Recall** ($C_{rec}$)**:** The ratio of correctly predicted contact frames to the total ground-truth contact frames.
  - **F1 Score** ($C_{F1}$)**:** The harmonic mean of precision and recall.

- **Contact Percentage** ($C_\%$)**:** The proportion of frames in the generated sequence where contact is detected. This reflects the intensity of the interaction.

- **Penetration Score** ($P_{hand}$)**:** To strictly evaluate physical violation, we utilize the precomputed Signed Distance Field (SDF) of the object meshes. For each frame $t$, we query the SDF value $d_i$ for every vertex $\mathbf{v}_i$ of the hand mesh. If $d_i < 0$, the vertex is inside the object. The penetration score is defined as the average summation of negative distances:

$$P_{hand} = \frac{1}{N_{frames}} \sum_{t=1}^{T} \sum_{i \in \mathcal{V}_{hand}} |\min(d_i(\mathbf{v}_{i,t}), 0)| \tag{7}$$

where $\mathcal{V}_{hand}$ denotes the set of hand vertices. The result is reported in centimeters (cm).

### B.5. Ground Truth (GT) Difference Metrics

These metrics measure the reconstruction accuracy by comparing the generated motion directly against the paired ground truth.

- **Position Errors** ($MPJPE, T_{root}, T_{obj}$)**:** We calculate the Mean Per-Joint Position Error (MPJPE) for body poses, and the global translation errors for the root joint ($T_{root}$) and the object ($T_{obj}$). All are measured as Euclidean distances in centimeters (cm).

- **Orientation Errors** ($O_{root}, O_{obj}$)**:** To measure rotational deviation, we calculate the error for the root orientation ($O_{root}$) and object orientation ($O_{obj}$). The error is formulated using the Frobenius norm of the difference between the predicted rotation matrix $R_{pred}$ and the ground truth matrix $R_{gt}$:

$$E_{rot} = \|R_{pred}R_{gt}^{-1} - \mathbf{I}\|_2 \tag{8}$$

where $\mathbf{I}$ is the identity matrix. This metric captures the geodesic distance in the $SO(3)$ manifold.

## C. HOI Motion and Condition Representation

### C.1. Human Motion Representation

We employ the SMPL-X parametric model to reconstruct the 3D human mesh from pose and shape parameters. Following the established protocol, the human motion feature $\mathbf{H} \in \mathbb{R}^{T \times 204}$ is defined as a 204-dimensional vector per frame, consisting of:

- **Joint Positions** ($24 \times 3$)**:** A flattened vector representing the 3D local positions of the 24 body joints.

- **Joint Rotations** ($22 \times 6$)**:** The 6D continuous rotation representations for the 22 major body joints (excluding hand articulations).

In addition to the pose parameters, the model utilizes a shape parameter vector $\boldsymbol{\beta} \in \mathbb{R}^{16}$ to control individual body morphology. Note that we simulate whole-body dynamics while focusing on gross manipulation, hence fine-grained finger articulations are omitted in this feature space to match the baseline dimensionality.

### C.2. Object Motion Representation

The object motion $\mathbf{O} \in \mathbb{R}^{T \times 12}$ is characterized by two primary components:

- **Global Translation** ($3$ **dims**)**:** The 3D centroid position of the object.

- **Relative Rotation** ($9$ **dims**)**:** The flattened $3 \times 3$ rotation matrix $R_{rel}(t)$.

The object vertices at frame $t$, denoted as $\mathcal{V}_t$, are derived from the canonical geometry $\mathcal{V}$ via the transformation: $\mathcal{V}_t = R_{rel}(t)\mathcal{V} + \mathbf{t}_{global}$.

### C.3. Condition and Masked Input Representation

To guide the diffusion process, we integrate multi-modal contextual cues following the CHOIS framework. The conditioning vector $\mathbf{c}$ is composed of:

- **Text Embedding:** Extracted using a frozen pre-trained CLIP text encoder.

- **Object Geometry:** Encoded via an MLP applied to the Basis Point Set (BPS) and broadcast temporally to all frames.

- **Sparse Waypoints:** Provided every 30 frames. Notably, only the object's $x$ and $y$ coordinates are provided at these keyframes (with the $z$-axis omitted) to simulate realistic 2D planning constraints.

**Masked Motion & Contact Indicators.** To explicitly model physical interactions, we augment the motion state with a **4-dimensional binary contact indicator**, specifying the contact state of hands and feet. Consequently, the network input is formulated as a masked motion sequence $\mathcal{M} \in \mathbb{R}^{T \times (12+204+4)}$. This sequence encodes the initial states (human pose + object pose at $t = 0$) and the sparse waypoint constraints, with all unconstrained intermediate frames masked (zero-padded) to be filled by the generative model.

## D. Model Architecture Details

Our framework consists of a Diffusion Backbone, Table 5 summarizes the key hyperparameters.

*Table 5.* Architecture Hyperparameters of MaMi-HOI.

| Parameter | Value |
|---|---|
| *Diffusion Transformer Backbone* | |
| Layers | 4 |
| Attention Heads | 4 |
| Latent Dimension ($d_{model}$) | 512 |
| Feedforward Dimension | 512 |
| Activation | GeLU |
| Dropout | 0.1 |
| Diffusion Steps | 1000 |

## E. Additional Diagnostic and Ablation Analyses

This section consolidates additional analyses conducted to further support the main claims. These results complement the main paper by providing dataset-wide feature-probing statistics, implementation clarifications for GAPA, efficiency measurements, module contribution analyses, downstream task evaluation, and a diagnostic study on generalization to unseen object geometries.

### E.1. Dataset-Wide Probe for Geometric Forgetting

To provide a more systematic diagnostic of Geometric Forgetting, we conduct a feature-probing experiment across the FullBodyManipulation test set. For each diffusion Transformer layer, we extract the hidden states and apply mean pooling over the temporal dimension. We then project all compared representations to matched dimensions and apply independent Layer Normalization to the hidden states, BPS geometry embeddings, and CLIP text embeddings before computing cosine similarity. This protocol isolates directional alignment and removes confounds caused by layer-wise magnitude shifts.

*Table 6.* Dataset-wide feature correlation probe on the FullBodyManipulation test set. Values report mean $\pm$ standard deviation. The baseline collapses to near-random geometric alignment in the deepest layer, whereas MaMi-HOI preserves a clear geometric signal.

| Layer Depth | Baseline | | MaMi-HOI | |
|---|---|---|---|---|
| | Semantic / Text | Geometry / BPS | Semantic / Text | Geometry / BPS |
| Layer_0 | $-0.023 \pm 0.047$ | $-0.013 \pm 0.048$ | $0.001 \pm 0.040$ | $0.045 \pm 0.066$ |
| Layer_1 | $-0.011 \pm 0.033$ | $-0.010 \pm 0.035$ | $-0.037 \pm 0.046$ | $0.074 \pm 0.079$ |
| Layer_2 | $-0.011 \pm 0.033$ | $0.018 \pm 0.040$ | $-0.067 \pm 0.046$ | $0.086 \pm 0.067$ |
| Layer_3 | $-0.048 \pm 0.027$ | $0.004 \pm 0.015$ | $-0.070 \pm 0.048$ | $0.092 \pm 0.067$ |

The baseline geometric alignment drops to $0.004 \pm 0.015$ in Layer_3, indicating that fine-grained BPS information is nearly absent from the deepest representation. In contrast, MaMi-HOI increases the geometric alignment across layers and reaches $0.092 \pm 0.067$ in Layer_3, supporting the claim that the proposed adapters preserve geometry-aware information inside the denoising network rather than only correcting the output coordinates.

**Random-vector calibration.** Because cosine values in a high-dimensional latent space can appear small in absolute magnitude, we further calibrate the probe by sampling pairs of uniformly distributed random unit vectors in the $D = 512$ latent space. The empirical random-vector cosine similarity has mean 0.00011 and standard deviation 0.04407. Thus, the baseline deep-layer geometric alignment of 0.004 falls within the random-noise range, while MaMi-HOI's deep-layer

geometric alignment of $0.092$ is more than $2\sigma$ above the random mean. This calibration suggests that a distinct geometric signal survives in MaMi-HOI's deep layers.

**Interpreting the sign of semantic correlation.** The dataset-wide semantic correlations are near zero and can be negative because different text prompts project semantic cues into different latent directions; averaging across prompts can therefore introduce sign cancellation. For feature-retention diagnostics, the cross-layer magnitude trend and the comparison to the random-vector baseline are more informative than the raw signed mean. Under this view, the baseline retains stronger semantic-directional signal than geometric signal in the deep layer, while MaMi-HOI prevents the geometric signal from collapsing to random noise.

### E.2. Why GAPA Differs from Distance-Based Regularization

A standard distance-based regularizer is imposed at the output level during training. It penalizes endpoint errors through the backward pass but does not explicitly provide the denoiser with local geometric context inside its intermediate representation. In contrast, GAPA is a structural inductive bias in the forward pass: it encodes dense BPS geometry and injects geometry-aware residuals through distance-aware cross-attention. The refinement can be summarized as

$$F_{\text{refined}} = F_{\text{mot}} + \Delta F_{\text{geo}}, \tag{9}$$

where $\Delta F_{\text{geo}}$ is computed from distance-biased attention between motion features and object geometry features. This design gives the denoising process direct access to local spatial context, allowing it to perform geometry-conditioned refinement rather than relying on a post-hoc pull from a loss term.

The probing results in Table 6 provide feature-level evidence for this distinction. If GAPA behaved merely as an output-level penalty, the deep hidden representations could still forget object geometry while only the final coordinates satisfy a contact loss. Instead, MaMi-HOI maintains a non-random deep-layer BPS correlation, indicating that geometric cues are preserved within the latent denoising manifold.

### E.3. GAPA Tensor Flow and Computational Cost

We further clarify the tensor flow of the BPS geometry representation. The raw BPS representation has shape $1024 \times 3$. The BPS coordinates for a mesh can be queried offline, and a lightweight MLP encodes them into a single $1 \times 256$ global geometry vector. This vector is then broadcast along the temporal axis to form $\hat{G} \in \mathbb{R}^{T \times 256}$, matching the notation used in the main method section.

Inside GAPA, the model still processes the full temporal sequence of $T$ frames. However, at each temporal frame, the spatial Key/Value token length for object geometry is one, because the dense 1024-point BPS signal has already been compressed into a single geometry token. Therefore, GAPA avoids dense point-cloud attention over 1024 spatial tokens and has $O(1)$ spatial complexity per frame with respect to the number of BPS basis points.

*Table 7.* Inference overhead breakdown. Runtime is reported as seconds per generated sequence, and VRAM is measured as peak GPU memory during inference.

| Model Configuration | Params (M) | Peak VRAM (GB) | Inference Time (s/seq) |
|---|---|---|---|
| Baseline (CHOIS) | 14.19 | 1.279 | 8.04 |
| + GAPA | 16.47 | 1.474 | 9.25 |
| + KHA | 25.52 | 1.251 | 10.51 |
| MaMi-HOI (Ours) | 27.80 | 1.437 | 11.15 |

The GAPA branch adds only a small number of parameters and moderate runtime overhead, because the dense BPS geometry is compressed before attention. The KHA branch introduces more parameters due to its trainable copy structure, but its memory footprint remains modest in our measurements because the injection layers are lightweight.

### E.4. Macro–Micro Module Contributions

We further isolate the roles of the two adapters. KHA primarily improves global trajectory adherence and whole-body kinematic coordination, whereas GAPA focuses on local contact and geometric alignment.

The single-adapter variants reveal the intended specialization: KHA better follows spatial waypoints, while GAPA better

*Table 8.* Macro–micro contribution analysis. KHA is stronger on trajectory adherence, while GAPA improves local contact metrics. The full model balances both objectives.

| Configuration | $T_s \downarrow$ | $T_{xy} \downarrow$ | $C_{prec} \uparrow$ | $C_{F1} \uparrow$ | Dominant Role |
|---|---|---|---|---|---|
| + KHA | **1.69** | **2.51** | 0.81 | 0.74 | Global routing / kinematics |
| + GAPA | 1.76 | 2.70 | **0.82** | **0.75** | Local geometry / contact |
| MaMi-HOI (Ours) | 1.53 | 2.14 | 0.80 | 0.73 | Balanced macro–micro synthesis |

enforces local contact. The full model may trade a small amount of isolated contact-score gain for global naturalness and trajectory accuracy, which is important for downstream execution.

### E.5. Downstream Task Success Evaluation

To assess practical utility, we evaluate generated sequences with a heuristic Task Success Rate (TSR) on macroscopic object transport. A sequence is counted as successful only when it satisfies both of the following conditions: (i) the final target-reaching error is less than 20 cm, and (ii) the hand-object distance during transport remains below 15 cm, indicating stable grasping or engagement without severe dropping or floating artifacts.

*Table 9.* Heuristic downstream task success evaluation. MaMi-HOI improves task execution reliability by combining global trajectory control with local contact preservation.

| Method | Heuristic TSR $\uparrow$ |
|---|---|
| Baseline | 63.69% |
| + KHA | 65.35% |
| + GAPA | 65.77% |
| MaMi-HOI (Ours) | **69.71%** |

The full model improves TSR by 6.02 percentage points over the baseline. This suggests that MaMi-HOI can provide more actionable motion priors for downstream simulation-based manipulation settings, where both target reaching and stable physical engagement are required.

### E.6. Generalization Behavior on Unseen 3D-FUTURE Objects

In the main paper, MaMi-HOI achieves stronger contact behavior on 3D-FUTURE, while its penetration score can be slightly higher than CHOIS on unseen shapes. To better interpret this trade-off, we measure the Average Hand-to-Object Surface Distance ($D_{hand}$) over active interaction frames.

*Table 10.* Average hand-to-object surface distance on 3D-FUTURE active frames. Lower values indicate tighter physical engagement with unseen object surfaces.

| Method | $D_{hand} \downarrow$ |
|---|---|
| CHOIS | 9.59 mm |
| MaMi-HOI (Ours) | **6.05 mm** |

CHOIS tends to avoid contact on challenging unseen shapes, which can reduce penetration but also increases hand-object distance. MaMi-HOI actively pulls the hands toward the unseen object surface, reducing $D_{hand}$ by approximately 37%. The slight millimeter-scale penetration increase is therefore a trade-off associated with tighter, more realistic grasping behavior under domain shift.

## F. Additional Qualitative Results

In this section, we provide additional HOI generation results across diverse settings, highlighting the versatility and effectiveness of our approach. As illustrated in the following figures, MaMi-HOI consistently generates high-quality motions with precise contacts and natural dynamics under various spatial constraints.

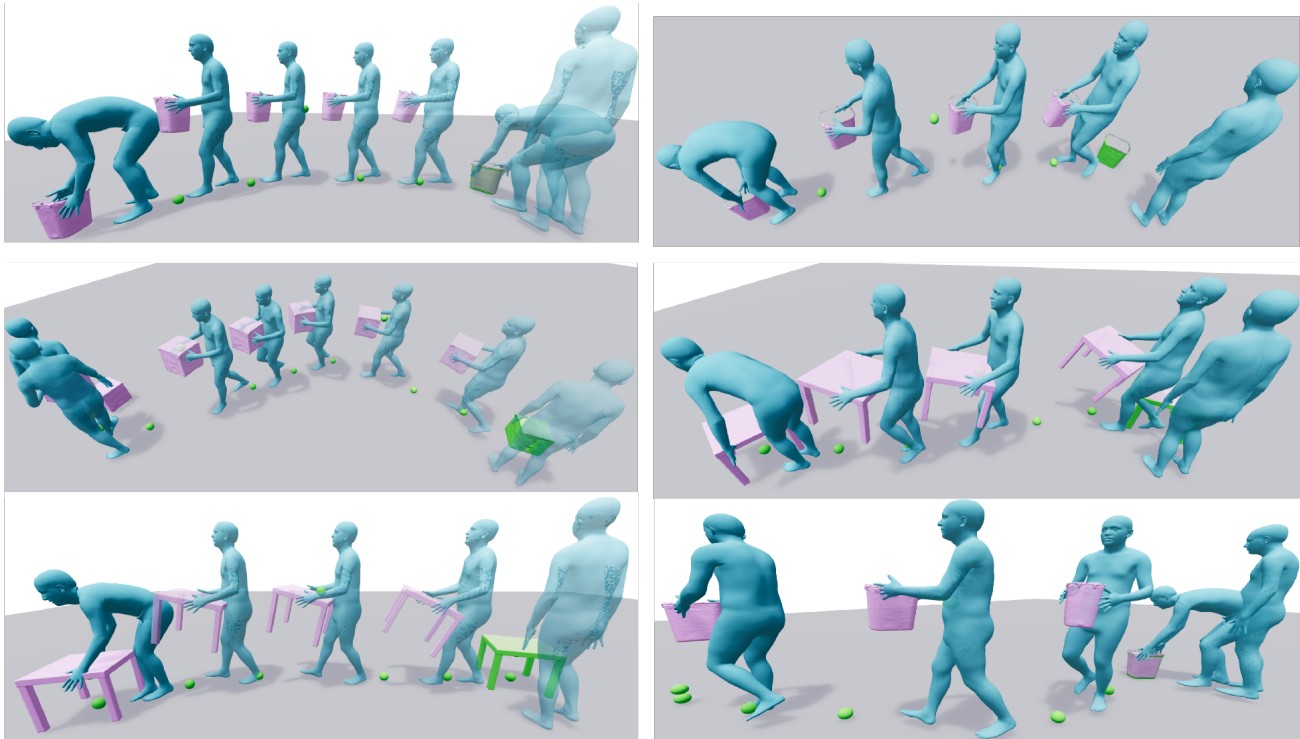

*Figure 7.* **Diverse Interactions with Complex Trajectories.** Visualization of MaMi-HOI generating sequences for various object categories. The results demonstrate the model's ability to synthesize coherent full-body motion and precise object manipulation even when following non-linear, curved spatial waypoints.

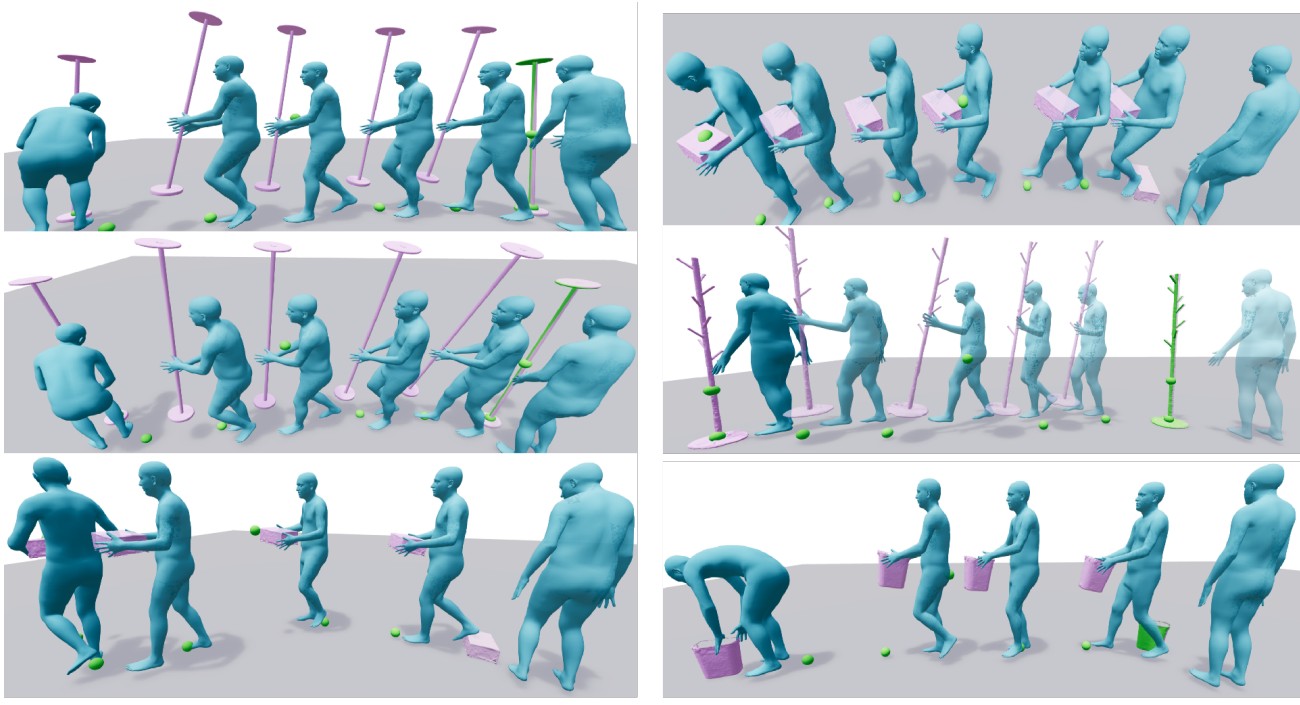

*Figure 8.* **Generalization to Challenging Geometries.** Additional qualitative results showcasing interactions with objects possessing complex topologies, such as tall floor lamps and coat racks. Despite the geometric complexity, MaMi-HOI accurately retrieves spatial cues to establish stable contacts and physically plausible dynamics for both lifting and dragging tasks.

