# OpenReview forum: "MaMi-HOI: Harmonizing Global Kinematics and Local Geometry for Human-Object Interaction Generation"
_ICML.cc/2026/Conference — ICML 2026 regular_

### Official Review · Reviewer_398i · 2026-02-16

**Soundness:** 3
**Presentation:** 3
**Significance:** 3
**Originality:** 2
**Overall Recommendation:** 4
**Confidence:** 3

**Summary:**

This paper studies 3D human–object interaction (HOI) generation with diffusion models under text, sparse object waypoints, and object geometry. It identifies an empirical failure mode termed Geometric Forgetting, where deeper diffusion transformer layers increasingly align with high-level semantic/kinematic intent while losing sensitivity to fine-grained object geometry, leading to contact errors despite correct global behavior. To address this, the authors propose MaMi-HOI, a dual-adapter framework that decouples macro-level kinematic coherence from micro-level geometric precision: (i) a Geometry-Aware Proximity Adapter (GAPA) injects dense geometry cues via distance-aware cross-attention for residual “snapping” corrections to improve contact, and (ii) a Kinematic Harmony Adapter (KHA) provides a parallel, spatially conditioned branch with zero-initialized injections to preserve natural whole-body dynamics while satisfying waypoint constraints. Experiments on FullBodyManipulation and generalization tests on 3D-FUTURE report improved waypoint adherence, motion quality, and interaction fidelity, and the method is demonstrated in a three-stage realistic 3D scene pipeline.

**Compliance With Llm Reviewing Policy:**

Affirmed.

**Final Justification:**

The paper identifies a concrete failure mode, "geometric forgetting", in diffusion-based HOI generation and proposes a dual-adapter solution. The rebuttal thoroughly addressed all technical concerns. Specifically, the efficiency analysis (Q2) confirmed GAPA's $\mathcal{O}(1)$ spatial complexity per frame, and the $\gamma$ ablation (Q3) justified the learnable penalty design. Regarding the follow-up on Q1, the authors clarified the apparent inconsistency between Figure 2 and Rebuttal Table 6 by attributing it to cross-prompt sign cancellation in dataset-wide averages. The follow-up on Q2 was also fully resolved. The primary remaining weakness is the limited architectural novelty of the individual components. However, the work's value lies in its principled problem diagnosis and validated system integration. I maintain my rating of 4.

**Key Questions For Authors:**

1) **Probe protocol for Geometric Forgetting.** How exactly are “hidden state” features and “condition” features defined and aggregated across tokens/frames/layers for the cosine similarity curves? If possible, control experiments (e.g., varying depth while matching representation size/normalization) to rule out confounds. A clearer, controlled probe would increase my confidence in the main motivation.
 2) **Efficiency and deployment cost.** What are the training/inference time and GPU memory costs compared to main baselines, and how much overhead comes from GAPA (1024-point BPS cross-attn) vs. KHA? In the 3D-scene pipeline, are BPS features precomputed per mesh or computed online?
 3) **$\gamma$ robustness.** How sensitive is performance to $\gamma$ initialization and training dynamics? Do you observe over-snapping vs. under-enforcement failure modes?
 4) **Generalization trade-off evidence on 3D-FUTURE.** The paper suggests higher penetration can arise from actively pursuing contact on unseen shapes. Can you provide additional evidence supporting this explanation?
 5) **Evaluator (HOI-BiGRU).** What data is the HOI-BiGRU evaluator trained on, and is there any overlap coupling with FullBodyManipulation/3D-FUTURE?

**Limitations:**

yes

**Strengths And Weaknesses:**

### **Strengths**
 1) **Technically solid and well validated.** The method is well-motivated by a concrete failure mode (contact precision degradation) and proposes a clear architectural intervention. Quantitative results across multiple axes (condition matching, motion quality, interaction quality, reconstruction) are generally consistent with the claimed benefits, and ablations (removing KHA / GAPA) support the complementary roles of the two adapters. Qualitative comparisons also align with the reported reductions in penetration/floating and improved whole-body coherence.
 2) **Clear decomposition.** The dual-adapter design is intuitive: KHA maintains global kinematic coherence under sparse constraints, while GAPA refines local contact using geometry-biased attention.
 3) **Generalization.** Evaluating on unseen object geometries (3D-FUTURE) is a relevant stress test for geometry-conditioned HOI synthesis, and the results suggest the approach retains benefits under domain shift.

### **Weaknesses**
 1) While the overall system is thoughtfully engineered and effective, key components resemble adaptations of established patterns: (i) KHA is closely related to common “control branch + zero-init residual injection” adapter mechanisms, and (ii) GAPA draws on prior geometric/vector attention ideas from point-based representations. The novelty is strongest in the diagnosis + system-level “retrieve-and-harmonize” integration for HOI, rather than a fundamentally new generative learning principle.
 2) GAPA introduces distance-aware cross-attention over BPS=1024 points, and the approach is used inside a diffusion process with many denoising steps. However, the paper provides a limited breakdown of training/inference time, memory overhead, and whether geometry features (BPS) are precomputed vs. computed online in the 3D scene pipeline. Given the paper’s emphasis on deployment in realistic 3D scenes, these missing details limit confidence in practicality.
 3) The learnable scalar $\gamma$ controls the strength of the distance penalty, causing overly aggressive snapping and kinematic artifacts (the paper briefly acknowledges potential limb over-stretching). The robustness of $\gamma$ across object scales/categories and its sensitivity to tuning are not thoroughly analyzed.
 4) Geometric forgetting evidence is underspecified. The probing setup needs clearer definition (which hidden states/conditions, how they are aggregated) and better control. As written, the phenomenon reads more like a compelling empirical observation than a well-isolated causal explanation.

---

> ### Author Rebuttal · Authors · 2026-03-31
>
> # Response to Reviewer 398i
> ## Question 1:
> >We thank the reviewer for this constructive feedback. To rule out confounds, we detail our exact aggregation protocol and present a strictly controlled experiment evaluated on the FullBodyManipulation test set.
> >
> >**1. Protocol & Normalization:** Hidden states are temporally aggregated via **Mean Pooling**. Crucially, we apply independent **Layer Normalization** to the matched dimensions of hidden states, BPS vectors, and Text embeddings. This guarantees cosine similarity measures *pure directional alignment*, eliminating layer-wise magnitude shift confounds.
> >
> >**2. Controlled Results:** Evaluated on the FullBodyManipulation test set.
> >
> >**Table 6: Controlled Probe (Baseline)**
> | Layer Depth | Semantic Alignment (Text) | Geometric Alignment (BPS) |
> | :--- | :--- | :--- |
> | Layer_0 | -0.023 ± 0.047 | -0.013 ± 0.048 |
> | Layer_1 | -0.011 ± 0.033 | -0.010 ± 0.035 |
> | Layer_2 | -0.011 ± 0.033 | 0.018 ± 0.040 |
> | **Layer_3 (Deep)**| **-0.048 ± 0.027** | **0.004 ± 0.015** |
> >
> >**Table 7: Controlled Probe (MaMi-HOI)**
> | Layer Depth | Semantic Alignment (Text) | Geometric Alignment (BPS) |
> | :--- | :--- | :--- |
> | Layer_0 | 0.001 ± 0.040 | 0.045 ± 0.066 |
> | Layer_1 | -0.037 ± 0.046 | 0.074 ± 0.079 |
> | Layer_2 | -0.067 ± 0.046 | 0.086 ± 0.067 |
> | **Layer_3 (Deep)**| **-0.070 ± 0.048** | **0.092 ± 0.067** |
> >
> >**3. Conclusion:** The baseline's geometric alignment drops to $0.004$ despite strict normalization, proving geometric forgetting is a genuine directional loss, not a scaling artifact. GAPA effectively counters this, increasing alignment to $0.092$.
> ## Question 2:
> >We thank the reviewer for raising these critical practical questions.
> >
> >**Table 8: Inference Overhead Breakdown**
> | Model Configuration | Params (M) | Peak VRAM (GB) | Inference Time (s/seq) |
> | :--- | :--- | :--- | :--- |
> | Baseline (CHOIS) | 14.19 | 1.279 | 8.04 |
> | + GAPA | 16.47 | 1.474 | 9.25 |
> | + KHA | 25.52 | 1.251 | 10.51 |
> | MaMi-HOI (Ours) | 27.80 | 1.437 | 11.15 |
> >
> >* KHA Overhead: Adds 11.3M parameters but **zero** peak VRAM overhead (1.251 vs 1.279 GB) due to memory-efficient linear layers.
> >* GAPA & BPS Efficiency: Raw 1024 x 3 BPS coordinates are queried offline. A fast online MLP (0.0358 ms) encodes them into a single $1 \times 256$ vector. Thus, GAPA's cross-attention operates on a sequence length of exactly 1, completely bypassing the quadratic bottleneck of dense point clouds and adding only minimal VRAM.
> ## Question 3:
> >**Table 9: Ablation on Penalty Strength ($\gamma$)**
> | Configuration | $P_{hand}$ (↓) | $C_{pre}$ (↑) | MPJPE (↓) | $T_{s}$ (↓) | $T_{e}$ (↓) | $T_{xy}$ (↓) |
> | :--- | :--- | :--- | :--- | :--- | :--- | :--- |
> | $\gamma = 0$ | 0.78 | 0.40 | 18.48 | 2.31 | 5.52 | 2.49 |
> |  5 | 0.76 | 0.36 | 29.56 | 1.84 | 5.08 | 2.34 |
> | 10 | 0.77 | 0.33 | 28.74 | 1.58 | 4.79 | 2.26 |
> |  Learnable (Ours) | **0.57** | **0.80** | **13.50** | **1.53** | **4.51** | **2.14** |
> >
> >Failure Modes: Fixed $\gamma=0$ causes *under-enforcement* (high penetration, $P_{hand}=0.78$). $\gamma=5$ and 10 causes *over-snapping*, severely distorting human pose.
> >
> >Decoupled Validation: While varying $\gamma$ heavily impacts local metrics and pose, global trajectory metrics ($T_s, T_e$) remain stable. This proves KHA robustly handles global navigation independent of GAPA's local geometry adjustments. A learnable $\gamma$ dynamically achieves the optimal balance.
>
> ## Question 4:
> >To quantify this, we introduce **Average Hand-to-Object Surface Distance ($D_{hand}$)** for active frames.
> >
> >**Table 10: Generalization Behavior on 3D-FUTURE**
> | Method | $D_{hand}$ (↓) |
> | :--- | :--- |
> | CHOIS | 9.59 mm |
> | **Ours** | **6.05 mm** |
> >
> >* Evidence: The baseline passively evades contact, hovering far from the object (9.59 mm) to artificially lower penetration. Our model actively pulls hands onto the unseen surface (6.05 mm, $\sim 37\%$ closer). Minor millimeter-scale penetration is the natural mathematical trade-off for achieving a tight, realistic grasp on completely novel topologies.
> ## Question 5:
> >We clarify the training data of the HOI-BiGRU evaluator to ensure no data leakage.
> >
> >* **Training Data:** To ensure fair comparison, we strictly followed the CHOIS baseline's evaluator design. The HOI-BiGRU evaluator is trained using a contrastive loss **exclusively on the training split** of the FullBodyManipulation dataset.
> >* **No Overlap/Coupling:** FullBodyManipulation uses a strict split (15 subjects for training, 2 for testing). All reported metrics are computed solely on the unseen test set. The evaluator has never seen the testing motions.
> >* **3D-FUTURE:** Evaluated on 17 unseen furniture models using motions strictly from the FullBodyManipulation *testing set*. >The evaluator has never encountered these novel geometries nor the associated testing motions during training.
> >
> >We will explicitly clarify this training split in the revised supplementary materials.

---

> > ### Author Rebuttal · Reviewer_398i · 2026-04-02
> >
> > Thank you for the detailed rebuttal. Q2–Q5 are resolved. Two clarifications are needed before I finalize my assessment:
> >
> > Q1 follow-up: Rebuttal Table 6 shows semantic alignment at Layer_3 is −0.048 after Layer Normalization — negative and near-zero across all layers. This contradicts Figure 2 in the paper, which shows semantic alignment increasing with depth. Was Figure 2 computed without Layer Normalization? If so, the "semantic overshadows geometric" narrative may be a scale artifact rather than a genuine directional effect. Please clarify.
> >
> > Q2 follow-up: Section 3.1 defines $\hat{G} \in \mathbb{R}^{T \times 256}$ as the geometry embedding used for contact modeling, while the rebuttal says GAPA's cross-attention uses sequence length = 1. Is BPS encoded as a single global vector and then broadcast to $T$ frames? Please confirm the exact mechanism.

---

> > > ### Author Response · Authors · 2026-04-03
> > >
> > > # Response to Q1 follow-up:
> > > > We sincerely thank the reviewer for their sharp observation. To clarify directly: **Both Figure 2 and Rebuttal Table 6 were computed under the exact same conditions, incorporating Layer Normalization.** We acknowledge the reviewer's concern regarding the -0.048 semantic alignment and the visual trajectory in Figure 2. This is not a contradiction, but a difference between a specific case and a global average subject to directional variance:
> > > >
> > > >**1. Figure 2 vs. Table 6:** Figure 2 illustrates the dynamic trajectory of a *single, specific interaction sequence*, where the semantic alignment happens to be strongly positive. Table 6, however, reports the average correlation across the *entire dataset*. Because different prompts project semantic features into entirely different sub-spaces, their Layer-Normalized correlations carry varying signs. Averaging these across the dataset introduces a cancellation effect, pulling the global mean closer to zero.
> > > >
> > > >**2. Assessing Feature Dominance via Signal Magnitude:** In the context of feature retention, the sign of the correlation is less critical than its absolute magnitude. In fact, Table 6 perfectly corroborates our "Geometric Forgetting" finding when analyzing the cross-layer trend. As network depth increases, the absolute magnitude of the geometric alignment drastically shrinks, ultimately collapsing to a near-zero floor(0.004) at Layer_3. Concurrently, the absolute magnitude of the semantic alignment grows (reaching |-0.048|), becoming significantly larger than the geometric alignment. This trend firmly proves that semantic features structurally outlast and overshadow delicate geometric cues in deep layers.
> > > >
> > > >**3. Camera-Ready Update:** To avoid any potential misinterpretation caused by presenting a single positive case, we will replace Figure 2 with the comprehensive dataset statistics (Table 6) in the camera-ready version to present a more balanced global view.
> > >
> > > # Response to Q2 follow-up:
> > > >We sincerely thank the reviewer for the question. **The reviewer’s intuition regarding the broadcast mechanism is correct, but we would like to clarify a misunderstanding regarding the module boundaries and the definition of "sequence length."**
> > > >
> > > >There is no contradiction. The confusion arises from conflating the **temporal sequence length** (number of frames, $T$) with the **spatial sequence length** (number of point/object tokens per frame, which is 1). Here is the exact chronological tensor flow to clarify the mechanism:
> > > >
> > > >**1. Global Encoding & Broadcasting (Prior to GAPA):**
> > > >The raw BPS representation ($1024 \times 3$) is first encoded by a fast online MLP into a single global geometry vector ($1 \times 256$). To support continuous dynamic interaction, this single vector is immediately **broadcasted along the temporal axis to $T$ frames**. This process happens early in the pipeline to form the geometry embedding $\hat{\mathcal{G}} \in \mathbb{R}^{T \times 256}$ (as defined in Section 3.1). Therefore, **before** the GAPA module even begins its operation, the object geometry is already formatted as a $T$-frame sequence.
> > > >
> > > >**2. T-Frame Processing Inside GAPA:**
> > > >The GAPA module takes this $T$-frame geometry embedding $\hat{\mathcal{G}}$ as its input alongside the $T$-frame human motion. GAPA explicitly models the dynamic interaction across all $T$ frames, ensuring continuous frame-by-frame contact alignment.
> > > >
> > > >**3. Clarifying "Sequence Length = 1" in the Previous Rebuttal:**
> > > >Inside GAPA, the Distance-Aware Cross-Attention computes the interaction frame-by-frame. When we stated "sequence length of exactly 1" in the previous rebuttal, we were exclusively referring to the **spatial sequence length of the Key/Value tokens at any single frame $t$**.
> > > In a standard dense point-cloud attention mechanism, the spatial sequence length per frame would be $N=1024$, causing an $\mathcal{O}(T \times 1024)$ bottleneck. Because our pipeline pre-compressed the 1024 points into a single feature vector per frame, GAPA's cross-attention physically operates on a **Key/Value spatial sequence length of exactly 1** at each step $t$.
> > > >
> > > >In summary, the geometry is broadcasted to $T$ frames *before* entering GAPA, allowing GAPA to process the full temporal sequence ($T$). The "1" solely refers to the compressed spatial token count per frame, reducing the spatial complexity to $\mathcal{O}(1)$. We will make these module boundaries and dimension definitions explicitly clear in the final manuscript.

---

### Official Review · Reviewer_JDnX · 2026-03-11

**Soundness:** 2
**Presentation:** 3
**Significance:** 2
**Originality:** 2
**Overall Recommendation:** 4
**Confidence:** 3

**Summary:**

This paper studies the problem of generating realistic 3D human–object interaction (HOI). The authors argue that existing diffusion-based HOI generation methods often struggle to maintain accurate contact between humans and objects, due to what they describe as “geometric forgetting,” where semantic features dominate over geometric information during generation. To address this issue, the paper proposes MaMi-HOI, a hierarchical framework that aims to combine global motion reasoning with local geometric constraints. The method introduces a macro-level module to model human motion kinematics and a micro-level module to refine spatial contact and geometry around interaction regions. Experiments are conducted on HOI generation benchmarks, with quantitative metrics and several qualitative examples used to evaluate interaction realism and contact accuracy.

**Compliance With Llm Reviewing Policy:**

Affirmed.

**Final Justification:**

Based on the above discussion, I tend to choose "weak accept".

**Key Questions For Authors:**

1. Visualization and qualitative evaluation: Could the authors provide more qualitative examples across a wider range of objects and interaction scenarios to better demonstrate the robustness and diversity of the generated HOI results?

2. Downstream applications: Do the authors have experiments showing how the generated interactions could benefit downstream tasks, such as simulation-based training, animation pipelines, or robotics applications?

3. Geometric forgetting analysis: The paper introduces the concept of geometric forgetting. Could the authors provide more quantitative evidence or diagnostic experiments that clearly demonstrate this phenomenon?

4. Module contribution: How much improvement does the micro-level geometric refinement bring compared to using only the macro-level motion generation module? A clearer ablation analysis would help clarify this.

**Limitations:**

yes

**Strengths And Weaknesses:**

Strengths:
1. The paper tackles an important problem in 3D vision and embodied AI. Generating realistic human–object interactions is relevant for applications such as simulation, robotics, and virtual content creation.

2. The motivation regarding the difficulty of maintaining accurate contact in HOI generation is reasonable. The idea of explicitly combining global motion modeling with local geometric refinement is intuitively appealing.

3. The method attempts to introduce a structured hierarchical design (macro-level motion modeling and micro-level geometric correction), which is a reasonable approach to addressing different scales of the HOI generation problem.

Weaknesses:
1. The evaluation mainly focuses on generation metrics and visual examples, but there is little evidence demonstrating the practical usefulness of the generated interactions. In particular, the paper does not provide downstream applications (e.g., robotics simulation, motion planning, or animation pipelines) that would demonstrate the value of the method beyond generation quality.

2. Some parts of the method description are not very easy to follow. The interaction between the macro-level and micro-level modules could be explained more clearly, and the pipeline figure could be simplified to better illustrate the data flow.

3. While the “geometric forgetting” phenomenon is an interesting observation, the empirical evidence supporting it is somewhat limited. The paper would benefit from a more systematic analysis showing how geometry information degrades across diffusion layers.

---

> ### Author Rebuttal · Authors · 2026-03-31
>
> # Response to Reviewer JDnX
> ## Question 1:
> >We sincerely thank the reviewer for this constructive suggestion. To better demonstrate the robustness and diversity of our generated human-object interactions (HOI) across a wider range of objects and complex scenarios, we have prepared an anonymous webpage with comprehensive video demonstrations and additional qualitative results：https://anonymous.4open.science/r/rebuttal-0777
> ## Question 2:
> >We sincerely thank the reviewer for this insightful suggestion. We demonstrate the downstream utility of our generated HOI sequences in two key areas:
>
> >**1. Robotics (Macroscopic Manipulation Task):** We evaluated our model on a macroscopic object transport task using the FullBodyManipulation dataset. Performance is measured by a **Heuristic Task Success Rate (TSR)**, where a sequence is a "Success" only if it meets two physical conditions:
> >* Target Reaching (< 20 cm error): Ensures effective open-loop macroscopic motion priors for embodied AI.
> >* Stable Grasping (< 15 cm hand-object distance): Ensures spatial engagement during transit, penalizing severe dropping or floating artifacts.
> >
> >**Table 3: Downstream Task Execution Evaluation**
> | Method | Heuristic Task Success Rate (TSR) $\uparrow$ |
> | :--- | :---: |
> | Baseline | 63.69% |
> | + KHA | 65.35% |
> | + GAPA | 65.77% |
> | **Ours (Full MaMi-HOI)** | **69.71%** |
> >
> >**Analysis**: Our full model achieves a 69.71% TSR (+6.02% over baseline), providing highly actionable motion priors without catastrophic error accumulation, improving sample efficiency for downstream reinforcement learning.
> >
> >**2. Animation Pipelines:** Current pipelines rely on generative models followed by slow, artifact-prone *post-hoc optimization* that degrades natural kinematics. By addressing geometric forgetting, our GAPA module injects geometric cues directly during the forward pass, generating motions natively ready for simulators without post-hoc fixes.
> ## Question 3:
> >We designed a rigorous feature-probing diagnostic experiment across the FullBodyManipulation test set to quantify this phenomenon.
> >1. Protocol: We extracted hidden states from each diffusion Transformer layer, applied Mean Pooling across the sequence dimension, and crucially, applied independent **Layer Normalization** to the features, BPS geometry vectors, and CLIP text embeddings before computing cosine similarity. This isolates pure directional alignment, regardless of latent space scale.
> >
> >2. Quantitative Evidence:
> >
> >**Table 4: Diagnostic Feature Correlation (Baseline vs. MaMi-HOI)**
> | Layer Depth | Baseline: Geometric Alignment (BPS) | Ours: Geometric Alignment (BPS) |
> | :--- | :--- | :--- |
> | Layer_0 | -0.013 ± 0.048 | 0.045 ± 0.066 |
> | Layer_1 | -0.010 ± 0.035 | 0.074 ± 0.079 |
> | Layer_2 | 0.018 ± 0.040 | 0.086 ± 0.067 |
> | **Layer_3 (Deep)**| **0.004 ± 0.015** | **0.092 ± 0.067** |
> >
> >**Analysis**: In the Baseline, geometric alignment collapses to near-zero noise by Layer_3, statistically confirming "geometric >forgetting." In contrast, MaMi-HOI steadily **increases** geometric alignment across layers, proving our paradigm successfully >reinjects geometric awareness.
> >
> ## Question 4:
> >To isolate the improvement brought by the micro-level geometric refinement (GAPA) versus the macro-level motion generation (KHA), we provide a comparison augmenting Table 5 with our TSR metric.
> *(Note: Please ensure module naming matches the main text).*
> **Table 5: Comparison of Two Modules**
> | Configuration | Trajectory $T_s$ (↓) | Trajectory $T_{xy}$ (↓) | Interaction $C_{prec}$ (↑) | Interaction $C_{F1}$ (↑) | Task Success (TSR) (↑) |
> | :--- | :--- | :--- | :--- | :--- | :--- |
> | Baseline | - | - | - | - | 63.69% |
> | + KHA | **1.69** | **2.51** | 0.81 | 0.74 | 65.35% |
> | + GAPA | 1.76 | 2.70 | **0.82** | **0.75** | **65.77%** |
> >
> >**Analysis:**
> >1.  Macro (+ KHA): Dominates global spatial routing, achieving superior trajectory metrics ($T_s$, $T_{xy}$).
> >2.  Micro (+ GAPA): Dominates local geometric contact. Used alone, it explicitly improves physical interaction ($C_{prec}$, $C_{F1}$).
> >3.  Task Impact: By resolving local spatial misalignments and unstable grasps, the micro-level correction alone yields a higher downstream TSR (65.77%) than the macro-level module (65.35%), confirming GAPA is indispensable for physically plausible manipulation.
> ## Weakness 2:
> >We sincerely thank the reviewer for this constructive feedback. We agree that clarifying the interaction between the macro-level and micro-level modules is important for readability. To address this, we have redesigned our pipeline figure to better illustrate the overall data flow and the precise interaction between these modules:https://anonymous.4open.science/r/rebuttal-0777/rebuttal_pipeline.png

---

> > ### Author Rebuttal · Reviewer_JDnX · 2026-04-03
> >
> > Thank you for your reply. My concerns have been addressed.

---

> > > ### Author Response · Authors · 2026-04-03
> > >
> > > Thank you very much for your time, effort, and constructive feedback throughout the review process. We are thrilled to hear that our rebuttal has fully addressed your concerns.
> > > As you kindly noted that your concerns have been resolved, we kindly wonder if you might consider adjusting your score accordingly, as suggested by the reviewing system. Your positive support would mean a great deal to our work.
> > > Thank you once again for your time and recognition!

---

### Official Review · Reviewer_A7r4 · 2026-03-15

**Soundness:** 2
**Presentation:** 3
**Significance:** 2
**Originality:** 2
**Overall Recommendation:** 4
**Confidence:** 4

**Summary:**

This paper addresses the problem that existing diffusion-based HOI models struggle to maintain precise spatial alignment, even when provided with explicit object waypoints. It proposes the perspective of "geometric forgetting", that the deeper layers in the network have features aligning better with semantics but worse with geometry. To address this problem, it proposes a retrieve-and-harmonize paradigm, which explicitly recovers lost geometric details to enforce contact precision while maintaining whole-body kinematics. Experiments show that the proposed method has better results in HOI, espsecially in long-term tasks with complex non-linear trajectories.

**Compliance With Llm Reviewing Policy:**

Affirmed.

**Final Justification:**

I appreciate the additional experimental results and analysis -- based on them, I am convinced that the "geometric forgetting" phenomenon is well-supported by evidences. And based on this, I think the overall story of the paper is quite interesting. So I increase my score to weak accept.

**Key Questions For Authors:**

Following the weaknesses above, I'm mostly curious about two things:
- Could there be more persuasive clues for the "geometric forgetting" phenomenon?
- How can we understand the reason *why* the GAPA module helps, beyond distance-based regularizations?

**Limitations:**

Yes, limitations are discussed in the paper.

**Strengths And Weaknesses:**

Strengths:
- The problem discussed in the paper, the balance between geometric constraints and kinematics, sounds reasonable for me.
- Experimentally, the proposed framework shows better performance than the baselines in most scenarios.

Weaknesses:
- I find it hard to tell whether "geometric forgetting" phenomenon is something statistically significant from the limited evidence in the paper. The only empirical evidence is Fig. 2, which only has 4 network layers, and it's a bit hard to tell the trend from these only 4 numbers. And how many data points are Fig.2 evaluated on? -- perhaps it'd be good to show the variance of each number as well? Additionally, it might also be good to have more explanations of how the feature cosine similarities are computed and normalized, as I feel different features may just have different scales or clusters in the latent spaces, and many factors can affect.
- I'm a bit unsure how exactly the GAPA module is helpful to the generations. I think instead of "adding back the geometric information" (that the model might have forgotten), it might be more possible that the explicit distance-based computations are enforcing the constraints (like contact).

---

> ### Author Rebuttal · Authors · 2026-03-31
>
> # Response to Reviewer A7r4
> ## Question 1: "Could there be more persuasive clues for the "geometric forgetting" phenomenon?"
> >
> >We sincerely thank the reviewer for this insightful question. Expanding upon the **probing experiment** conducted in our original >study, we now present its comprehensive, dataset-wide results to provide definitive quantitative evidence for the "geometric >forgetting" phenomenon.
> >
> >By systematically measuring the feature correlation (normalized cosine similarity) across the entire FullBodyManipulation test set >for both the Baseline and our full MaMi-HOI model, we statistically confirm the existence of this bottleneck and clearly demonstrate >the effectiveness of our solution.
> >
> >**Table 1: Probing Feature Correlation in Baseline Model**
> >
> >| Layer Depth | Semantic Alignment (Text) | Geometric Alignment (BPS) |
> >| :--- | :--- | :--- |
> >| Layer_0 | -0.023 ± 0.047 | -0.013 ± 0.048 |
> >| Layer_1 | -0.011 ± 0.033 | -0.010 ± 0.035 |
> >| Layer_2 | -0.011 ± 0.033 | 0.018 ± 0.040 |
> >| **Layer_3 (Deep)**| **-0.048 ± 0.027** | **0.004 ± 0.015** |
> >
> >
> >**Table 2: Probing Feature Correlation in MaMi-HOI (Ours)**
> >
> >| Layer Depth | Semantic Alignment (Text) | Geometric Alignment (BPS) |
> >| :--- | :--- | :--- |
> >| Layer_0 | 0.001 ± 0.040 | 0.045 ± 0.066 |
> >| Layer_1 | -0.037 ± 0.046 | 0.074 ± 0.079 |
> >| Layer_2 | -0.067 ± 0.046 | 0.086 ± 0.067 |
> >| **Layer_3 (Deep)**| **-0.070 ± 0.048** | **0.092 ± 0.067** |
> >
> >**Key Persuasive Clues:**
> >* **Proof of the Phenomenon:** In the Baseline (Table 1), as the network deepens to Layer_3, the geometric alignment collapses >to near-zero noise (0.004 ± 0.015), even though semantic alignment remains active. This probing statistically confirms that >standard diffusion layers indeed selectively "forget" fine-grained object geometry.
> >* **Proof of the Solution:** In stark contrast, when probing our MaMi-HOI framework (Table 2), the geometric alignment not only >survives but steadily **increases** across the layers (reaching 0.092 ± 0.067 at Layer_3).
> This comprehensive probing experiment provides concrete evidence that "geometric forgetting" is a statistically significant bottleneck, directly justifying the necessity of our design paradigm. We will include these tables and the probing details in the revised manuscript.
> ## Question 2: "How can we understand the reason why the GAPA module helps, beyond distance-based regularizations?"
> >We highly appreciate the reviewer’s deep dive into the underlying mechanism of our method. It is a crucial distinction: GAPA is fundamentally different from, and far superior to, simple distance-based regularizations. We explain this from both a theoretical/architectural perspective and through empirical feature-level evidence.
> >
> >**1. Theoretical Mechanism: Forward Feature Injection vs. Backward Loss Penalty**
> >A standard distance-based regularization operates exclusively at the output level during training (as a loss function). It forces the network to blindly adjust its weights via gradients to minimize endpoint errors, without providing the network with explicit geometric context about *what* it is interacting with.
> >
> >In contrast, GAPA is a structural inductive bias operating in the **forward pass**. It encodes the dense 1024-point BPS representation and utilizes a distance-aware cross-attention mechanism to explicitly compute a geometric residual. This residual is directly injected into the motion latent space:
> >$$F_{refined} = F_{mot} + \Delta F_{geo}$$
> >Instead of a blind "pull" from a loss function, GAPA explicitly equips the network with local geometric context (curvatures, surface normals implied by BPS) directly within the intermediate feature manifold, enabling geometry-conditioned denoising.
> >
> >**2. Empirical Proof: Feature-Level Correlation (The Probing Experiment)**
> >If GAPA were merely acting as a regularization equivalent, the deep hidden layers would still suffer from "geometric forgetting," merely outputting coordinates that satisfy the loss. However, our comprehensive probing experiment (detailed in the previous response, Table 2) proves otherwise.
> >
> >In the baseline model, the deep layers lose geometric awareness (correlation drops to `0.004`). With GAPA, the geometric correlation in Layer_3 actively rises to `0.092`. This is concrete evidence that GAPA physically preserves and harmonizes the high-dimensional geometric semantics *inside* the network's latent representation, rather than just imposing a spatial penalty at the end.

---

> > ### Author Rebuttal · Reviewer_A7r4 · 2026-04-03
> >
> > Thanks for the detailed responses! My concerns are partially resolved. I appreciate the more systematical evaluation of the feature similarities across the dataset, but I have some remaining concerns:
> > - The consine similarity at the magnitude of $10^{-2}$ seems a bit too close to zero for all features. Usually, for CLIP-like features with explicit losses enforcing cross-modality relations, the cosine similarity could be $>0.2$. On one hand, the change from $0.004$ to $0.092$ is indeed a positive sign; but on the other hand, it may still be a bit hard for me to tell whether this is significant enough.
> > - Could the authors also calculate or test, for two random unit vectors of the latent dimension, what is the average cosine similarity between them, as a calibration to these numbers in the table?

---

> > > ### Author Response · Authors · 2026-04-04
> > >
> > > # Response to Reviewer A7r4
> > > >We appreciate the reviewer’s rigorous suggestion to calibrate the latent space. Computing the expected similarity of random vectors is indeed a highly appropriate method to contextualize the absolute magnitudes in a high-dimensional space.
> > > >Following your suggestion, we computed the cosine similarity between pairs of uniformly distributed random unit vectors in our $D=512$ latent space. The empirical calibration yields:
> > > >* **Mean Cosine Similarity:** 0.00011
> > > >* **Standard Deviation ($\sigma$):** $\pm$ 0.04407
> > > >
> > > >**1. Verifying "Geometric Forgetting":**
> > > >In the Baseline model, the geometric alignment in the deep layer (Layer_3) drops to **0.004**.  Statistically, this indicates that the geometric signal in the deep layers has fallen completely within the variance of random noise. The feature is practically lost, verifying the "forgetting" phenomenon.
> > > >
> > > >**2. The Significance of our 0.092 Result:**
> > > >In our MaMi-HOI model, the deep layer geometric alignment is **0.092**. We agree with the reviewer that this is not a massive absolute value. However, evaluated against the baseline, it represents a shift to **$>2\sigma$** above the random noise mean. In a 512-dimensional space, a $>2\sigma$ deviation statistically confirms that a distinct and persistent geometric signal has survived the network depth and is preserved outside the noise distribution.
> > > >
> > > >**3. Regarding the Comparison to CLIP (>0.2):**
> > > >We respectfully note that comparing these hidden state values directly to CLIP features involves fundamentally different optimization landscapes. CLIP is trained explicitly via contrastive loss (e.g., InfoNCE), whose direct mathematical objective is to force matched multi-modal vectors to be highly parallel, naturally yielding similarities >0.2.
> > > >
> > > >In contrast, our diffusion backbone minimizes a denoising MSE loss. The hidden states interact with the geometric conditions via cross-attention (using dot products to compute attention weights for feature aggregation) rather than explicit contrastive alignment. Therefore, the network is never mathematically forced to strictly align the *angles* (cosine similarity) of its hidden states with the condition vectors.
> > > >
> > > >Given this non-contrastive mechanism, we do not expect CLIP-level cosine similarities. The fact that a structurally persistent geometric signal (0.092, $>2\sigma$) manages to survive in the deep layers of MaMi-HOI—whereas it decays into pure noise (0.004) in the baseline—is the core evidence that our architectural design successfully maintains spatial awareness.

---

### Decision · Program_Chairs · 2026-04-30

**Decision:**

Accept (regular)

**Comment:**

This paper received mixed negative and positive reviews.
After checking all reviews and author responses, AC recommends accepting this paper.
Although `Reviewer A7r4` is negative to this paper, the reviewer did not provide response to the authors' last response.
AC checked and authors addressed the concerns properly.
It is recommended to include valuable points raised during the rebuttal period.